# Modelling water isotopologues ($^1H^2H^{16}O$, $^1H_2^{17}O$) in the coupled numerical climate model iLOVECLIM (version 1.1.5)

Thomas Extier[1], Thibaut Caley[1], Didier M. Roche[2,3]

[1] Univ. Bordeaux, CNRS, Bordeaux INP, EPOC, UMR 5805, F-33600 Pessac, France.

[2] Laboratoire des Sciences du Climat et de l'Environnement, LSCE/IPSL, CEA-CNRS-UVSQ, Université Paris-Saclay, 91191 Gif-sur-Yvette, France.

[3] Earth and Climate Cluster, Faculty of Sciences, Vrije Universiteit Amsterdam, De Boelelaan 1085, 101 HV Amsterdam.

*Correspondence to*: Thomas Extier (thomas.extier@u-bordeaux.fr)

**Abstract.**

Stable water isotopes are used to infer changes in the hydrological cycle for different climate periods and various climatic archives. Following previous developments of $\delta^{18}O$ in the coupled climate model of intermediate complexity iLOVECLIM, we present here the implementation of the $^1H^2H^{16}O$ and $^1H_2^{17}O$ water isotopes in the different components of this model, and calculate the associated secondary markers d-excess and $^{17}O$-excess in the atmosphere and ocean. So far, the latter was modelled only by the atmospheric model LMDZ4. Results of a 5,000 years equilibrium simulation under preindustrial

conditions are analysed and compared to observations and several isotope-enabled models for the atmosphere and the ocean components.

In the atmospheric component, the model correctly reproduces the first order global distribution of the $\delta^2H$ and d-excess as observed in the data (R=0.56 for $\delta^2H$ and 0.36 for d-excess), even if local differences are observed. The model-data correlation is within the range of other water isotope-enabled General Circulation Models. The main isotopic effects and the latitudinal

gradient are properly modelled similarly to previous water isotope-enabled General Circulation Models simulations despite a simplified atmospheric component in iLOVECLIM. One exception is observed in Antarctica where the model does not correctly estimate the water isotope composition, consequence of the non-conservative behaviour of the advection scheme at very low moisture content. The modelled $^{17}O$-excess presents a too important dispersion of the values in comparison to the observations and is not correctly reproduced in the model mainly because of the complex processes involved in the $^{17}O$-excess

isotopic value. For the ocean, the model simulates adequate isotopic ratio in comparison to the observations, except for local areas such as in the surface Arabian Sea, a part of the Arctic and West equatorial Indian ocean. Data-model evaluation also presents a good match for the $\delta^2H$ over the entire water column in the Atlantic Ocean, reflecting the influence of the different water masses.

## 1 Introduction

Stable water isotopologues ($^1H^2H^{16}O$, $^1H_2^{16}O$, $^1H_2^{17}O$, $^1H_2^{18}O$) expressed hereafter in the usual delta notation with respect to V-SMOW scale (Dansgaard, 1964) are important tracers of the hydrological cycle and are measured in a large variety of archives to reconstruct climate variations. At first order, the $\delta^2H$ and $\delta^{18}O$ of precipitation measured in polar ice cores can be used as a proxy of past temperature at the drilling site (e.g. Johnsen et al., 1972; Lorius et al., 1979; Jouzel, 2003). As they present the same variations, we can derive a second-order parameter called deuterium excess (d-excess) from the difference

between the $\delta^2H$ and $\delta^{18}O$. During evaporation, kinetic non-equilibrium processes affect the relationship between oxygen and hydrogen isotopes and lead to a deviation from the global Meteoric Water Line, which represents the linear relationship between $\delta^2H$ and $\delta^{18}O$ (Craig, 1961; Dansgaard, 1964):

$$d-\text{excess} = \delta\,^2H - 8 \times \delta^{18}O \tag{1}$$


This parameter is a classical polar ice-core tracer that can be used to provide additional constraints on past climates and changes in the atmospheric water cycle. The deuterium excess is conventionally interpreted in terms of temperature at the moisture source, or shifts in moisture origin (Stenni et al., 2001; Vimeux et al., 2002; Masson-Delmotte et al., 2005) even if it can also be impacted by local temperature (Masson-Delmotte et al., 2008a) and by mixing along trajectory (Hendricks et al., 2000;

Sodemann et al., 2008). Modelling studies such as Risi et al. (2013) also suggested that the d-excess is controlled by convective processes and rain re-evaporation at the tropics and by effect of distillation and mixing between vapors from different origins at high latitudes. Recently, Landais et al. (2021) also shown using the first 800,000 years d-excess record that precipitation, seasonality and moisture source regions changes in the past can complicate the interpretation of the d-excess.

Following experimental developments for an accurate measurement of $^1H_2^{17}O$ abundance (Barkan and Luz, 2007; Landais et al., 2008), a second-order parameter, the $^{17}O$-excess, has been defined such as:

$$^{17}O-\text{excess} = \ln\left(\frac{\delta^{17}O}{1000} + 1\right) - 0.528 \times \ln\left(\frac{\delta^{18}O}{1000} + 1\right) \tag{2}$$

The $^{17}O$-excess is then multiplied by $10^6$ and expressed in per meg since magnitudes are very small (Landais et al., 2008). Note that we used the logarithm notation for $^{17}O$-excess following Luz and Barkan (2005). This definition makes it very sensitive to mixing between vapors of different origins (Risi et al., 2010).

The $^{17}O$-excess is commonly used in ice core based paleoclimate studies to give information on the relative humidity over the

ocean (e.g. Landais et al., 2008, 2018; Risi et al., 2010; Steig et al., 2021). This proxy is controlled by kinetic fractionation during evaporation, and similarly to d-excess, is very sensitive to empirical parameters determining the supersaturation in polar

clouds (Landais et al., 2012; Winkler et al., 2012). Since influences of temperature or condensation altitude on [17]O-excess are expected to be insignificant in contrast to d-excess, measurements of [17]O-excess have an added value with respect to d-excess and can be used to disentangle the parameters (temperature, relative humidity) that affect the water isotopic composition. For example, Risi et al. (2010) shown that the different behaviors of d-excess and [17]O-excess in polar regions could be related to fractionation processes along the distillation pathway from the evaporative source to polar region, that affect more the d-excess than the [17]O-excess, with [17]O-excess recording more the signal from low latitudes during surface evaporation. Modelling the [17]O-excess is still very challenging since it depends on complex processes that have to be properly reproduced in the climate models. To date, only the LMDZ4 model has included the [17]O-excess (Risi et al., 2013). However, even if the processes that control the [17]O-excess are more complex than those controlling the d-excess, the combination of the d-excess, [17]O-excess and [18]O could bring new information on the understanding of past changes in local temperature, moisture origin and conditions at the moisture source.

Among the new proxies to document the water isotopic ratio in precipitation, the hydrogen isotope composition of plant wax (alkanes) has been found to reflect predominantly local continental rainfall fluctuations (e.g. Schefuß et al., 2005; Collins et al., 2013; Kuechler et al., 2013). Isotopic changes are primarily controlled by moisture loss by evapotranspiration, soil water conditions and precipitations rates, but the vegetation and isotopic enrichment effects are also to consider (Hou et al., 2008; Sachse et al., 2012; Kahmen et al., 2013a,b). Another method has also been developed to extract the fossil water (fluid inclusions) of speleothem records (Vonhof et al., 2006; van Breukelen et al., 2008). It then becomes possible to realize hydrogen and oxygen stable isotope analyses of fossil precipitation waters and to document the deuterium excess values in the past, outside the limited region of ice core presence.

Similarly to continental records, the isotopologues in ocean surface waters track regional freshwater balance and then the hydrological cycle (Craig and Gordon, 1965). Water isotopologues in seawater can therefore be used as a proxy for salinity since surface freshwater exchanges are important in determining the variability of both variables. Seawater oxygen isotope concentration preserved in carbonate from organisms such as foraminifera allows qualitative estimations of past regional changes in salinity and ocean circulation (Schmidt et al., 2007; Caley et al., 2011). It has been suggested that combining seawater hydrogen isotopes ($\delta^2$H obtained from alkenones or other biomarkers) with oxygen isotope ($\delta^{18}$O obtained from zooplankton calcite shells of foraminifera) could be a promising way to quantitatively estimate salinity variability (Rohling, 2007; Legrande and Schmidt, 2011, Leduc et al., 2013; Caley and Roche, 2015).

With the emergence of new paleoproxy to document water isotopologues in atmospheric and oceanic components of the climate system, the need to develop and use isotope-enabled models, and in particular coupled ocean-atmosphere models, as never been greater (e.g. Schmidt et al., 2007; Tindall et al., 2009; Werner et al., 2016; Cauquoin et al., 2019a). These later allow more complex assumptions related to paleoclimatic proxies to be examined (LeGrande and Schmidt, 2006; Schmidt et

al., 2007). For example, the simulation of the climate and its associated isotopic signal can provide a "transfer function" between the isotopic signal and the considered climate variable such as precipitation rate/water isotopes in precipitation or salinity/water isotopes in seawater relationships.

Since the initial work of Joussaume et al. (1984) and Jouzel et al. (1987), much progress has been done in atmospheric general circulation models (AGCMs) (e.g. Hoffmann et al., 1998; Noone and Simmonds, 2002; Mathieu et al., 2002, Risi et al., 2010; Werner et al., 2011) that can accurately simulate the $\delta^{18}O$ of precipitation. The subsequent development of water isotopes modules in oceanic general circulation models (OGCMs) (Schmidt, 1998; Delaygue et al., 2000; Xu et al., 2012) opens the possibility for coupled simulations of present and past climates, conserving water isotopes through the hydrosphere (Schmidt
et al., 2007; Zhou et al., 2008; Tindall et al., 2009; Werner et al., 2016; Cauquoin et al., 2019a). General Circulation Models (GCMs) have first been used to simulate separately water isotopes in the atmospheric and oceanic components but are now capable of running snapshot coupled simulations with the water isotope-enabled. Running transient coupled simulations like the last deglaciation or the Holocene remains however still challenging due to high computing cost of these GCMs. Given the computing resources needed to run coupled climate models, applying intermediate complexity coupled climate models with
water isotopes like iLOVECLIM to long-term palaeoclimate perspectives is suitable (e.g. Caley et al., 2014). Other isotope-enabled intermediate complexity models exist like CLIMBER (Roche et al., 2004), or fast GCM like SPEEDY-IER (Dee et al., 2015), that could be used to improve our understanding of the relationship between water isotopologues, of second-order parameter (like d-excess) and of the climate over a broad range of simulated climate changes.

Oxygen isotopes (18, 16) have been implemented in iLOVECLIM, allowing fully coupled atmosphere-ocean simulations. The detailed implementation of oxygen isotopes in iLOVECLIM and the evaluation against observed data in water samples and carbonates can be found in Roche (2013), Roche and Caley (2013) and Caley and Roche (2013). In the present manuscript, we present the design and the validation of $\delta^2H$ water isotopes as well as deuterium excess and $^{17}O$-excess in the coupled climate model iLOVECLIM for the atmospheric and oceanic components. The agreements and differences from the direct
comparison between modelling results under preindustrial conditions with (1) multiple datasets and (2) several isotope-enabled GCMs results for the atmosphere and the ocean components will be discussed to determine the potential and the interest of using iLOVECLIM for paleoclimatic studies.

**2 Description of the water isotopic scheme in iLOVECLIM**

**2.1 Atmospheric component ECBilt**

The iLOVECLIM model (version 1.1.5) is a derivative of the LOVECLIM-1.2 climate model extensively described in Goosse et al. (2010). It is composed of an atmospheric, oceanic, land surface and vegetation component. The atmospheric component ECBilt is a quasi-geostrophic model with a T21 spectral grid (resolution of 5.6° in latitude and longitude) with a complete

description of the water cycle from evaporation, condensation to precipitation. The timestep of the atmospheric component is 6 hours. It is subdivided in three vertical layers: (1) between the surface and 650 hPa, (2) between 650 and 350 hPa, and (3) between 350 and 0 hPa. 800, 500 and 200 hPa are respectively the mid-point of each layer. The humidity is contained only in the first layer and representative of the total humidity content of the atmosphere. Evaporative water fluxes are added to this humid layer and vertical advection is computed. Water fluxes crossing the limit between the humid and dry layers are rained out instantly as convective rain. For specific humidity of the humid layer larger than 80 % (set as the saturation humidity at given temperature), the excess water is removed as large-scale precipitation. If large-scale precipitation occurs with negative temperatures, excess precipitation is removed as large-scale snowfall.

With regards to water isotopes, the main development lies in the atmospheric component in which evaporation, condensation and the existence of different phases (liquid and solid) all affect the isotopic conditions of the water isotopes. The methodology used to trace the hydrogen water isotopes in ECBilt is identical to the description made in Roche (2013) for the oxygen water isotopes. We used the same equations presented for the $^{18}$O in Roche (2013) but with adapted fractionation coefficients for the hydrogen and for $^{17}$O. We present in this section the equations for the heavy/light isotope ratios. Additional information on general water scheme formulation can be found in Roche (2013).

In ECBilt, the water isotopic quantity is expressed as a single tracer of water and the humidity is assumed to be only in the first layer. For $^1H^2H^{16}O$ / $^1H_2^{16}O$, it is defined as a function of the quantity of precipitable water for the whole atmospheric column ($\tilde{q}$ which depends on the mass of the water, the surface area of the cell and the water density) and of the ratio ($R^H$) between the number of moles of $^1H^2H^{16}O$ and the number of moles of $^1H_2^{16}O$:

$$\tilde{q}^H = \tilde{q} \times R^H \tag{3}$$

The isotopic ratio then changes within the water cycle, from evaporation to precipitation. The evaporation term for hydrogen water isotopes cannot be simply written like for the humidity because there is no vertical discretization for water isotopes in the model. The solution adopted by Roche (2013) is to compute the water isotopic ratio in the evaporation using a Craig and Gordon (1965) type-model in the formulation adapted by Cappa et al. (2003). The hydrogen isotopic ratio of evaporating moisture can then be written as:

$$R_E^H = \alpha_{diff}^* \left( \frac{R_{eq}^H - h_a^* R_a^H}{1 - h_a^*} \right) \tag{4}$$

where $R^H_{eq}$ is the isotopic ratio at equilibrium with the ocean, $R^H_a$ the isotopic ratio of the humidity in the atmosphere and $h_a^*$ is an apparent relative humidity value for the atmosphere. $\alpha^*_{diff}$ is a ratio of molecular diffusivity and defined for the hydrogen such as:

$$\alpha_{diff}^{*} = \left(\frac{D^{H}}{D}\right)^{n} \tag{5}$$

with $D^{H}$ the molecular diffusivity of water $^{1}H^{2}H^{16}O$, D the molecular diffusivity of water $^{1}H_{2}^{16}O$ and n a coefficient that varies with turbulence and evaporative surface (Brutsaert, 1975; Mathieu and Bariac, 1996). The molecular diffusivity ratio for $^{1}H^{2}H^{16}O$ / $^{1}H_{2}^{16}O$ is set to 0.9755 (Merlivat, 1978) and 0.9855 for $^{1}H_{2}^{17}O$ / $^{1}H_{2}^{16}O$ (Barkan and Luz, 2007).

Since ECBilt only includes three layers, it is supposed that precipitation always forms in isotopic equilibrium with the surrounding moisture with instantaneous rainout to the surface. The convective precipitations, large-scale precipitation and snow are in equilibrium with isotopic values (using temperature at 650, 800 and 650 hPa respectively). When computing the precipitation and snow fractionation coefficients (see Roche, 2013), we take into account the temperature, the equilibrium fractionation coefficients between the different water phases for the hydrogen (Merlivat and Jouzel, 1979) and the ratio of hydrogen isotopes in vapor. In these equations, the hydrogen equilibrium fractionation coefficient between liquid water and vapor is taken from Majoube (1971a) and depends on the temperature:

$$\alpha_{l-v}^{H} = \exp\left(\frac{24844}{T^{2}} - \frac{76.248}{T} + 0.052612\right) \tag{6}$$

For $^{17}O$, the fractionation between liquid water and vapor is calculated from Majoube (1971a), Barkan and Luz (2005; 2007):

$$\alpha_{l-v}^{O} = \exp\left(\frac{1137}{T^{2}} - \frac{0.4156}{T} - 0.0020667\right) \times 0.529 \tag{7}$$

The equilibrium fractionation coefficient between solid water and water vapor for hydrogen is taken from Merlivat and Nief (1967) and depends on the temperature as well:

$$\alpha_{s-v}^{H} = \exp\left(\frac{16289}{T^{2}} - 0.0945\right) \tag{8}$$

For $^{17}O$, the fractionation between solid water and vapor is calculated from Majoube (1971b), Barkan and Luz (2005; 2007):

$$\alpha_{s-v}^{O} = \exp\left(\frac{11.839}{T^{2}} - 0.028224\right) \times 0.528 \tag{9}$$

## 2.2 Ocean and land surface components

The oceanic component CLIO has a 3x3° horizontal resolution, 20 vertical layers and a free surface. All the variables are calculated with a daily timestep. In the ocean, the water isotopes are mass conserving and act as passive tracers under equilibrium fractionation ignoring the small fractionation implied by the presence of sea-ice (Craig and Gordon, 1965).

For the land surface model, the isotopes water implementation in the bucket follows the same procedure as for the water. If re-evaporation occurs on land, it is assumed to be at equilibrium (without fractionation). A snow layer is also taken into account. Above a given threshold, the isotopic water and snow contents in the soil and snow buckets are routed to the ocean without fractionation.

## 2.3 Simulation setup

We present results of a 5,000 years equilibrium run under fixed preindustrial boundary conditions. The atmospheric $pCO_2$ is set to 280 ppm, methane concentration is 760 ppb and nitrous oxide concentration is 270 ppb. The orbital configuration is calculated from Berger (1978) with constant year 1950. We use present-day land sea mask, freshwater routing and interactive vegetation. With regards to the water isotopes, the atmospheric moisture is initialized at 0 and the $\delta^2H$ at 0 ‰. The consistency of our integration is checked by ensuring that the water isotopes are fully conserved in our coupled system. The model has

been run at T21 spatial resolution and the outputs are computed with an annual timestep.

To investigate the seasonal variations of the model in comparison to the observations, and to estimate the range/dispersion of the modelled results, we performed a 100 years simulation starting from the equilibrium run, with monthly outputs for the climate and the isotopes. This simulation is investigated in Section 3.1.4.

## 2.4 Observational data and water isotope-enabled GCMs

To allow for comparison and discussion with iLOVECLIM results, global hydrogen and d-excess isotopic datasets for the atmosphere from the Global Network of Isotopes in Precipitation (GNIP) dataset (IAEA, 2023) and Masson-Delmotte et al. (2008b) have been used. The original GNIP dataset has been subsampled to keep only the stations where the isotopic composition has been reported for a minimum of 3 calendar years within the period 1961-2008. To evaluate the seasonal evolution of the model, we looked at the evolution of precipitations and atmospheric isotopic ratio at several locations

distributed on multiple continents to reflect the variety of climate: Pretoria (25.73°S, 28.18°E), Belem (1.43°S, 48.48°W), Ankara (39.95°N, 32.88°E) and Reykjavik (64.13°N, 21.92°W). Present day measurements of $\delta^{17}O$ and $^{17}O$-excess from multiple studies (Landais et al., 2008, 2010, 2012; Luz and Barkan, 2010; Uemura et al., 2010; Winkler et al., 2012; Pang et al., 2015; Tian et al., 2021; IAEA, 2023) have been used. Note that the data of Uemura et al. (2010) are for the vapor and not the precipitation and does not allow for a direct model-data comparison.

The GISS global seawater isotope database (Schmidt et al., 1999) has been used to compare the $\delta^2$H and d-excess with the ocean component in the model. We looked at the surface distribution of the isotopes for the first oceanic layer at 5 m depth in the model and selected GISS sea water values between 0 and 10 m to be representative of the surface.

To evaluate our model results against water isotope-enabled GCMs, we used several model outputs: ECHAM5-wiso (Steiger et al., 2018), GISS (Schmidt et al., 2007), LMDZ4 (Risi et al., 2010, Risi et al., 2013), MIROC (Kurita et al., 2011), CAM

(Lee et al., 2007) and MPI-ESM-wiso (Cauquoin et al., 2020). The GISS, LMDZ4, MIROC and CAM data are from the Stable Water Isotope Intercomparison Group Phase 2 (SWING2) (Risi et al., 2012). $\delta^2$H$_{seawater}$ in MPI-ESM-wiso has been calculated from $\delta^{18}$O$_{seawater}$ and d-excess outputs.

## 3 Results and discussion

### 3.1 Water isotopic composition in the atmosphere

#### 3.1.1 Annual $\delta^2$H$_{precipitation}$

The annual-mean modelled distribution of $\delta^2$H$_{precipitation}$ is presented in comparison to observations on Fig. 1a. The latitudinal gradient from the poles to the equator is correctly reproduced in the model with low values at high latitudes (cold and dry regions) and high values at lower latitudes. Regions like central Africa and northern region of South America show however differences with the data since the modelled $\delta^2$H$_{precipitation}$ is underestimated in comparison to the few measurements available.

This could be due to one of the well-known iLOVECLIM biases that is the overestimation of the precipitation in these regions. The west coast of South America also presents discrepancies between the model and the GNIP data (Fig. 1a). This could be related to the coarse model resolution that may not perfectly reproduce the observed $\delta^2$H$_{precipitation}$ since the value is representative of a larger area. Finally, the modelled $\delta^2$H$_{precipitation}$ over northern America and Europe is higher than the observations. The difference in atmospheric isotopic ratio of precipitation over land and ocean is however well reproduced in

the model with values closed to zero over the Pacific, Atlantic and Indian oceans, and values lower than -50 ‰ and -80 ‰ respectively over the Arctic and Austral oceans (Fig. 1a).

We also compared the zonal distribution of several water isotope-enabled GCMs for results that co-locate with observations. From mid to low latitudes, all models show similar $\delta^2$H$_{precipitation}$ with iLOVECLIM being higher than the other GCMs below

20°S and above 30°N. Despite these biases, iLOVECLIM reproduces the global trend of low values at high latitudes and high values at low latitudes, as observed in the data (Fig. 2a). At high latitudes, iLOVECLIM models an isotopic ratio that is too high compared to the one in ECHAM5-wiso, GISS, LMDZ4, MIROC and CAM models, as well as in the GNIP data with values between up to -453 ‰ (Fig. 2a). These very low measured values over Antarctica can be explained by the low temperature (with a continental effect) and by other influences like moisture transport or the distance from the coast that add

complexity in modelling this region. Since iLOVECLIM only have three vertical layers in comparison to the 19 to 26 vertical layers for the other GCMs, we cannot properly reproduce the isotopic variations at these latitudes as a consequence of the non-

conservative behaviour of the advection scheme at very low moisture content. However, no model is able to correctly reproduce these very low values as observed in the measurements. All the GCMs model higher values, between -305 ‰ and -365 ‰.

In order to further evaluate our model results against water isotope-enabled models and the observations, we analysed the standard deviation (SD), correlation (R) and root mean square error (RMSE), combined in a Taylor Diagram (Fig. 3). In all these figures, we removed Antarctic values for the reason explained above. We observe for the $\delta^2H_{precipitation}$ that ECHAM5-wiso is the model that has the best correlation coefficient with the observation (R=0.64 vs R=0.56 for iLOVECLIM – Fig. 3a). The different GCMs have close correlation coefficient (between 0.59 and 0.64), standard deviation (between 40.21 and 46.43)

and RMSE (between 34.94 and 39.82). The iLOVECLIM model presents a lower standard deviation (SD=29.93) and RMSE than the other models (Fig. 3a). However, considering the close metrics between all models, iLOVECLIM presents the advantage to run faster than other GCMs and is perfectly justified for the use of long-term global climate simulation.

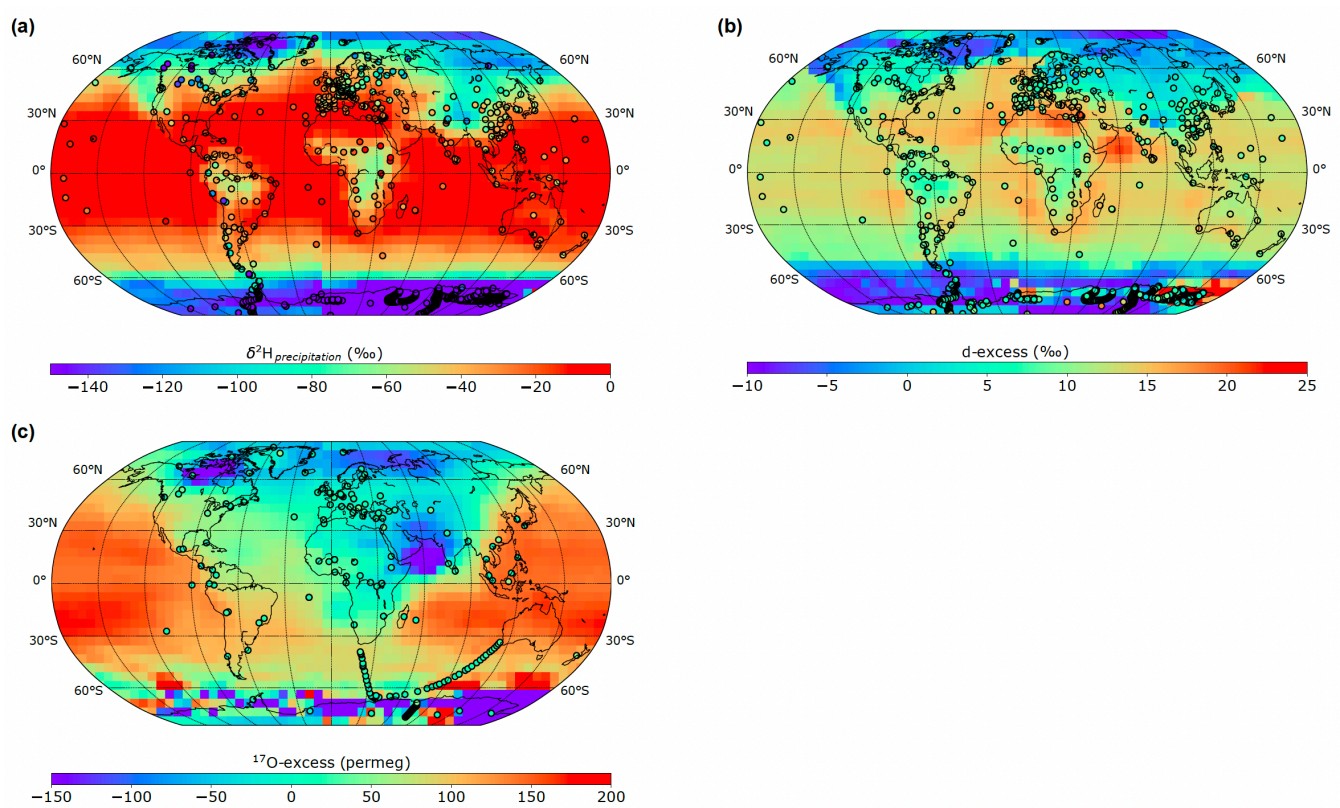

Figure 1: Model-data evaluation of the annual-mean isotope distributions. (a) $\delta^2H$ in precipitation, (b) d-excess and (c) $^{17}O$-excess in iLOVECLIM. The model results are compared to observations (in circles).


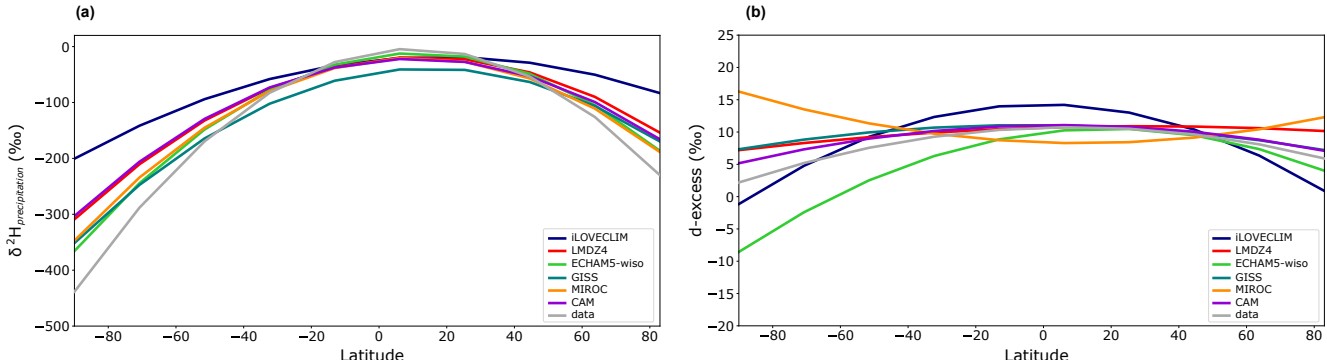

**Figure 2: Multi-model zonal (a) δ²H_precipitation and (b) d-excess comparison. The model results (in color) are compared to observations (in grey). The different lines are polynomial regression curves for the model results that co-locate with the observations.**

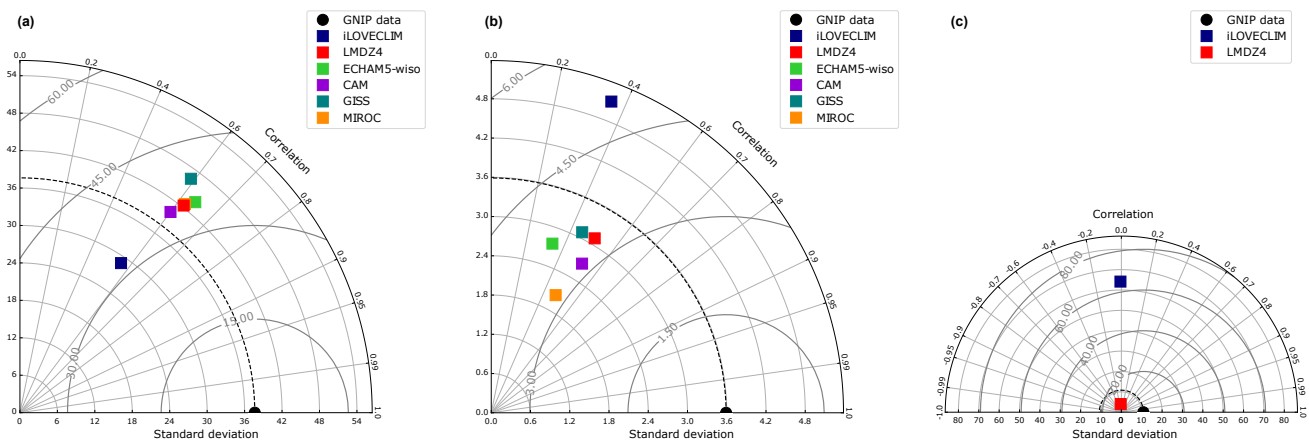

**Figure 3: Taylor diagram representing (a) δ²H_precipitation, (b) d-excess and (c) ¹⁷O-excess values for different climate models (iLOVECLIM, LMDZ4, ECHAM5-wiso, CAM, GISS and MIROC) without Antarctic values. The simulated values are plotted against the observations. The dotted curved line indicates the reference line (standard deviation of the observations) and the bold grey contours represent RMSE values.**

The linear relationship between $\delta^{18}O$ and $\delta^2H$ ($\delta^2H = 8*\delta^{18}O + 10$) established by Craig (1961) and defined as the global Meteorological Water Line can also be verified in the model. The model values match the GNIP observations and correctly reproduces the linear trend between the $\delta^{18}O$ and $\delta^2H$ of precipitation with a correlation coefficient of 0.99.

### 3.1.2 Annual deuterium excess

The annual-mean d-excess distribution is derived from the oxygen and hydrogen isotopic ratio. To evaluate the accuracy of
the model, we compare the model results to the observations. As observed for the $\delta^2H_{precipitation}$, the d-excess presents a

latitudinal gradient with low negative values to the poles and high positive values to the equator (Fig. 1b and Fig. 2b). The modelled values fit well with the observations at global scale. Differences between the model and the observations remain for some regions like over India where the modelled d-excess is slightly higher than the observations. More generally, iLOVECLIM models too high d-excess values from mid to low latitudes (Fig. 2b). The modelled d-excess over Greenland, and especially the coastal areas, is negative whereas the few available data points indicate positive values that are up to 20 ‰ higher. Similarly to the annual $\delta^2H_{precipitation}$ distribution, the d-excess over Antarctica is not correctly reproduced in the model and presents outliers values in the coastal regions. The local data show values between 5 and 10 ‰ whereas the model calculates values ranging from -10 to 25 ‰ or higher in the region of Adélie Land (Fig. 1b). In Figure 2b, we excluded these outlier values for a more suitable model intercomparison. Zonal mean d-excess values from mid to high latitudes modelled by LMDZ4, GISS, and CAM are too high compared to the observations, whereas values from ECHAM5-wiso are systematically too low. The MIROC model is the only one that shows a different trend in the zonal distribution of the d-excess, with higher values in the high latitudes and low values to the equator. Over the ocean, few d-excess data points are available but the model presents an overall good agreement with the GNIP data with mean values ranging from -10 ‰ over the Arctic and Austal oceans to 17 ‰ over the Atlantic and Pacific oceans. A maximum in d-excess is reach over the Arabian sea with 20.6 ‰.

In comparison to the measurements for the atmosphere, iLOVECLIM has a correlation coefficient that is within the range of others models (0.34 to 0.52), but with a higher SD compared to the observations and other GCMs. The CAM model has the best correlation coefficient with the observations whereas LMDZ4 has the closest standard deviation relative to the observations (Fig. 3b). Within all models, MIROC is the one with the lowest SD and RMSE. However, considering the general low correlation coefficient for all models, they all do not perfectly reproduce the d-excess variations as observed in the data. iLOVECLIM however presents the advantage to run faster than the other GCMs. The same caution should be required for iLOVECLIM as for other GCMs when investigating past changes in d-excess

The relationship between the d-excess and the $\delta^2H_{precipitation}$ can be investigated and shows that it is partially driven by high latitudes values, mainly in Antarctica, as presented in Fig. 4. From the globally available data, a relationship between d-excess and $\delta^2H_{precipitation}$ exists with high d-excess value (~15 ‰) for very low $\delta^2H_{precipitation}$ values (around -400 and 0 ‰), whereas lower d-excess is observed for mean $\delta^2H_{precipitation}$ between -250 and -300 ‰. The low $\delta^2H_{precipitation}$ values correspond to high latitudes values, mostly corresponding to Antarctic values, that drive the relationship between d-excess and $\delta^2H_{precipitation}$ ($R^2$=0.50 when considering all values, $R^2$=0.10 for values without the high latitudes). Similar relationship between the d-excess and $\delta^2H_{precipitation}$ is observed in the iLOVECLIM model. Highest d-excess values are obtained for low $\delta^2H_{precipitation}$ values (around -200 ‰) and lower d-excess for intermediate $\delta^2H_{precipitation}$ (Fig. 4). The shape of the regression curves is however different between the data and the model because of outlier modelled d-excess values that are too high in the model. These data points mainly correspond to Antarctic values as already observed on Fig. 1.

Antarctic isotopic values are not computed correctly due to issues in the conservation of water in the advection scheme at very
low humidity content, a fact that was already highlighted in Roche (2013). Improving the conservation in the spectral advection
scheme is beyond the scope of the present study. We thus removed these Antarctic values in the following to investigate the
isotopic trend without the influence of this region. This results in a better agreement between the data and iLOVECLIM model
(with a correlation coefficient of 0.71), even if differences are observed with generally lower d-excess value in the model than
in the data for low $\delta^2 H_{\text{precipitation}}$ (Fig. 4).

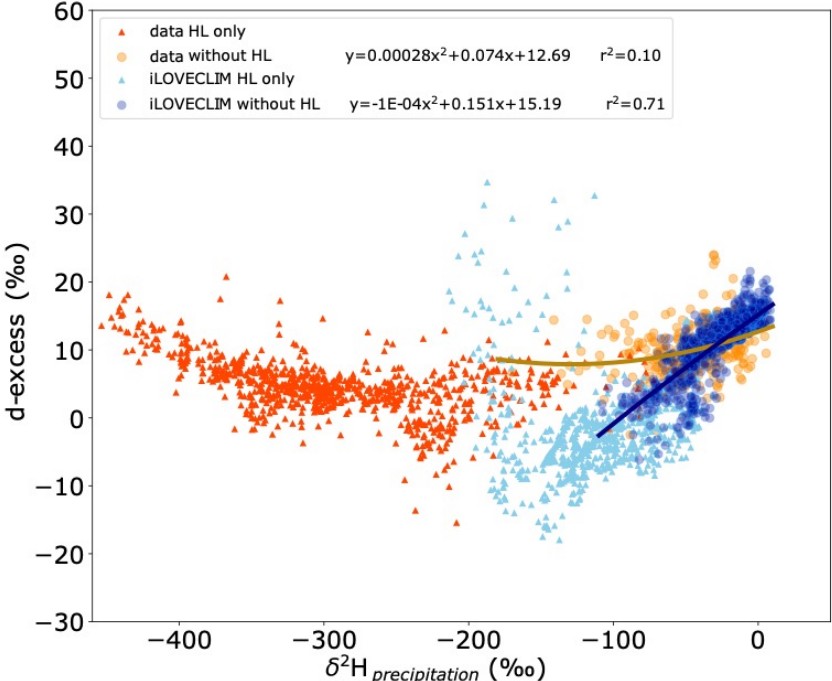


**Figure 4: Global relationship between d-excess and $\delta^2 H$ in precipitation. High latitude values (above 60°N and below 60°S) are
presented with the red triangles for the data and with the light blue triangles for iLOVECLIM. Data for other regions are presented
with the orange circles for the measurements and with the dark blue circles for the model. Regression curves for the data and the
model, without high latitudes values, are also shown in orange and dark blue.**


For the d-excess, the range of modelled values can be large for some locations (as already seen in Fig. 1). Thus, we can evaluate
the ability of the model to reproduce the d-excess in comparison to the observed data, as presented in Fig. 5. The distribution
of most d-excess values is centred around values between 8-18 ‰. Low correlation coefficient is obtained due to outlier d-
excess values, but statistical significance between the model and the data is obtained with a p-value of 3e-4 (<0.001). This
attests of a good representation of the d-excess in the model (excluding Antarctic values). This is also supported by the
modelled d-excess in LMDZ4 that presents similar values than in iLOVECLIM (Fig. 5). However, considering the larger
dispersion of the values in our model compared to LMDZ4 and to the fact that the uncertainties on the d-excess measurements
are large, the relationship between model and data might vary and get closer to the expected 1:1 line.

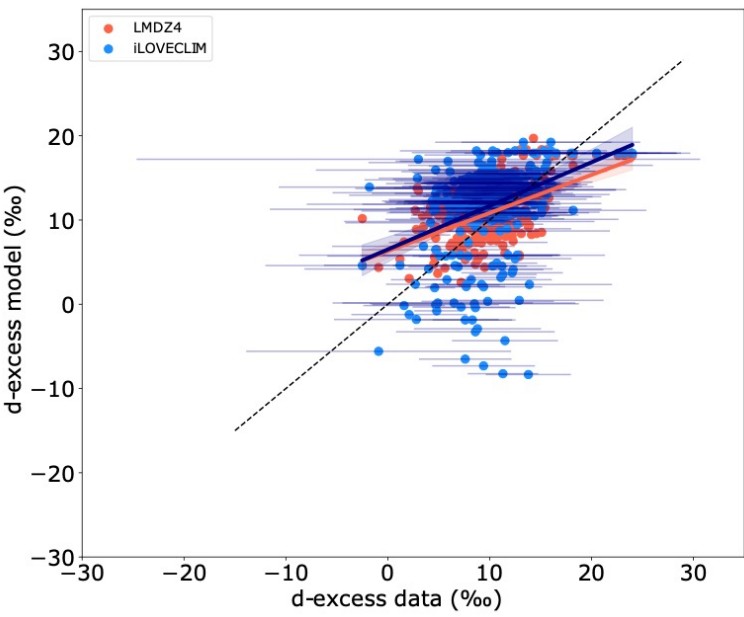

**Figure 5: Relationship between the modelled d-excess in iLOVECLIM (blue) and in LMDZ4 (red) versus measurements without Antarctic values. The errors bars associated with the data are shown at 2σ. The 1:1 line is shown with the black dashed line. Regression lines for iLOVECLIM and LMDZ4 are in dark blue and red respectively with the confidence bands.**

### 3.1.3 $^{17}$O-excess distribution

Modelled $^{17}$O-excess shares common pattern with $\delta^{17}$O (itself presenting the same spatial pattern than $\delta^{18}$O, see Appendix A) with low values over the high latitudes of the Northern Hemisphere and higher values over land (Fig. 1c). The $^{17}$O-excess presents values between 0 and 100 permeg over the Atlantic Ocean, that are lower than in the Indian and Pacific oceans. In comparison to the LMDZ4 model that is currently the only GCM to include the $^{1}$H$_2$$^{17}$O (Risi et al., 2013), iLOVECLIM presents higher values for most of the latitudes, due to these high values over the ocean. The latitudinal gradient is also larger than in LMDZ4 that has relatively homogenous values between 70°S and 90°N. The model reproduces $^{17}$O-excess values that are close to observations over North America, Europe and Africa (Fig. 1c). But $^{17}$O-excess over the Arabian Sea and northern Canada has probably too negative values. Similarly to d-excess and due to the outlined problem in modelling this region, the $^{17}$O-excess modelled over Antarctica present a wide range of values from high negative to high positive and does not fit with ice core measurements.

Comparison can be done between model and observations for the $^{17}$O-excess (Fig. 6a). A wide dispersion of the $^{17}$O-excess values (excluding values in Antarctica) is observed in the model, but statistically significant with a p-value of 0.041 ($<0.05$). Higher values than observations are modelled from mid to low latitudes and lower values than observations at high latitudes of the northern hemisphere (Fig. 6b). $^{17}$O-excess has been previously modelled in LMDZ4 (Risi et al., 2013), with a lower

dispersion of the values than iLOVECLIM but no clear trend as expected from the data (Fig. 6a). We observe for the [17]O-excess a low negative correlation coefficient for iLOVECLIM and LMDZ4 with respect to observations. Interestingly, the opposite pattern in the models compared to observations suggests that the physical processes at play are not fully understood and require further investigation. The standard deviation and root mean square error is better for LMDZ4 than for iLOVECLIM (Fig. 3c), suggesting that our model does not correctly reproduce the [17]O-excess and has a too important dispersion of the

values.

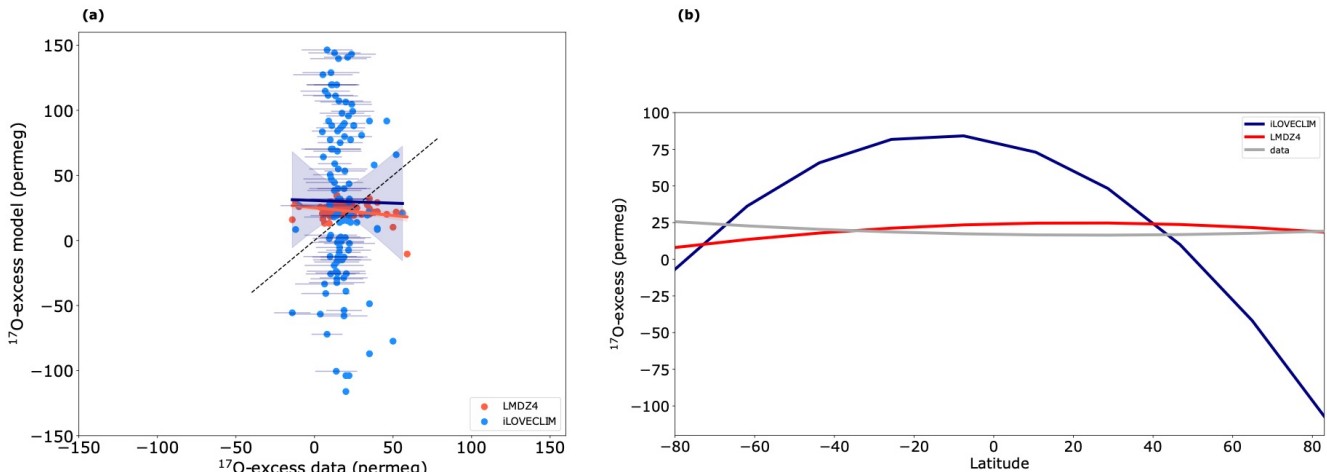

**Figure 6: (a) Relationship between the iLOVECLIM modelled isotopic value and [17]O-excess measurements, without values in Antarctica. LMDZ4 model results are also presented. The regression curves between model and data are presented in dark blue for**
**iLOVECLIM and red for LMDZ4 with the confidence bands. The 1:1 line is shown with the black dashed line. The errors bars associated with the data are shown at 1σ. (b) Zonal [17]O-excess comparison. The model results (in color) are compared to observations (in grey). The different lines are polynomial regression curves for the model results that co-locate with the observations.**

### 3.1.4 Seasonal variations

We compare the seasonal model results for precipitation, $\delta^2 H_{precipitation}$, d-excess and [17]O-excess to the GNIP monthly data at several locations representative of various climate conditions to have a global overview: South Africa (Pretoria), South America (Belem), eastern Mediterranean (Ankara) and northern Atlantic (Reykjavik). [17]O-excess values are presented only for Ankara and Reykjavik, since no data are available for the other stations. We extracted the model results at the corresponding locations but due to the coarse resolution of the model, regional biases exist as depicted in previous section. We performed a
mean over the last 10 years of the simulation and normalized the results (we subtracted the annual-mean and divided by the standard deviation for each station) for easier comparison with the data. The seasonal evolution of precipitation and isotopic ratio in the model is then not expected to perfectly reflect the measurements. We then present the normalized values for both model and GNIP data.


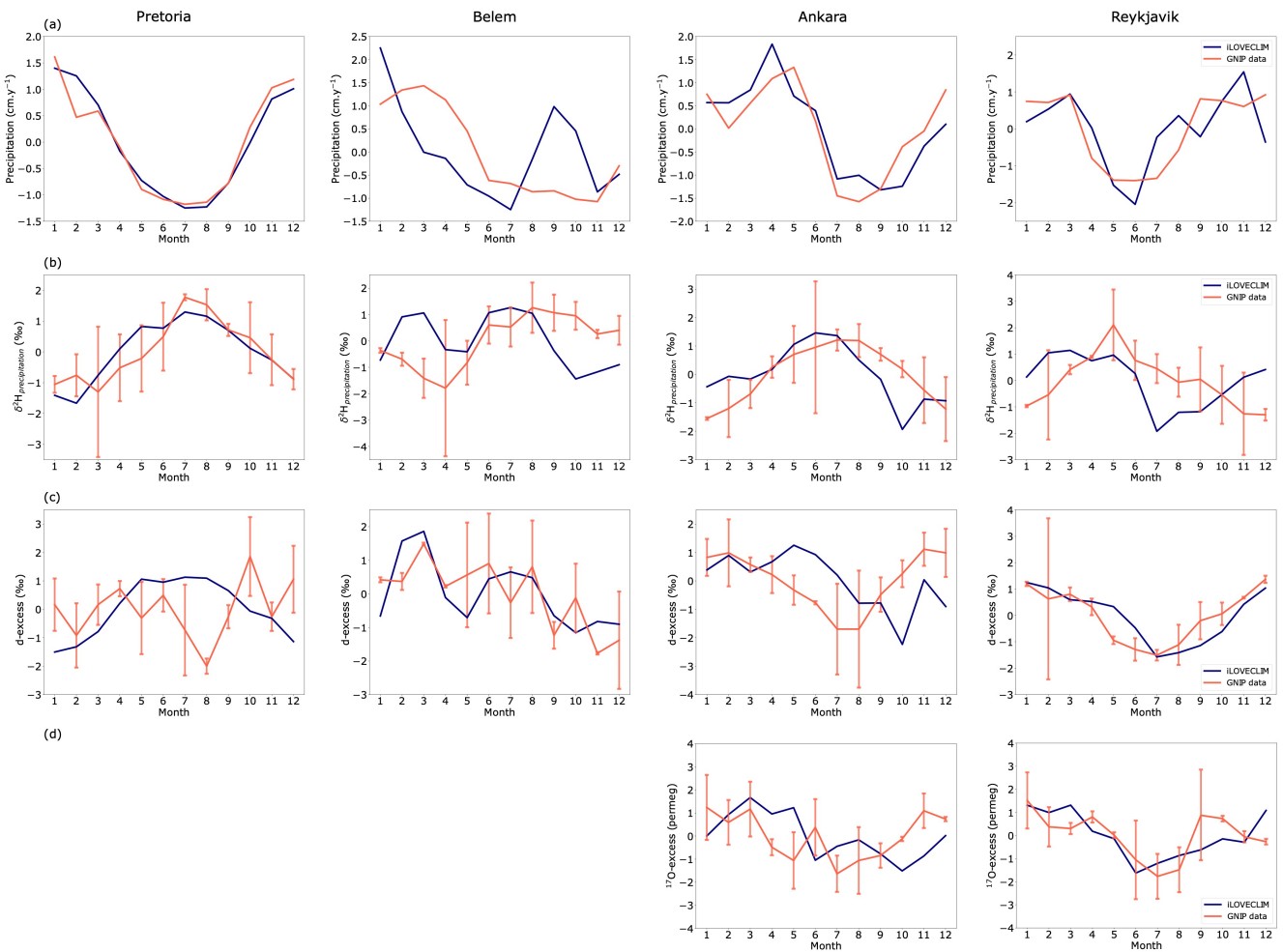

**Figure 7: Monthly evolution of (a) precipitation, (b) $\delta^2H_{precipitation}$, (c) d-excess and (d) $^{17}O$-excess at several stations (different columns for Pretoria, Belem, Ankara and Reykjavik). The red line is the GNIP data measured at the station and the blue line is the iLOVECLIM model result at the corresponding location. The data and model results have been normalized. The error bars for the**
**data are also shown at 2σ.**

There is a good agreement in precipitation at Pretoria and Ankara between the observation and the model that correctly reproduce the seasonal cycle (Fig. 7a). For Belem and Reykjavik stations the model shows some differences, namely higher precipitations in September and October at Belem and higher monthly amplitude at Reykjavik. Good correlation is observed

for the modelled $\delta^2H_{precipitation}$ in comparison to observations at Pretoria and Ankara (even if the October value is very low). As for precipitations, the amplitude of $\delta^2H_{precipitation}$ variations is different between the model and the data at Belem and Reykjavik (Fig. 7b). But the overall model behaviour in reproducing seasonal variations of $\delta^2H_{precipitation}$ can be validated based on these

observations, especially when considering that the uncertainties associated with the data can be as large as the measurement itself. The d-excess variations show however larger differences between the model and the observations. The modelled d-excess at Reykjavik shows a good agreement with the observation, while larger amplitude of the variations is observed at Belem (Fig. 7c). At Ankara, the modelled d-excess is delayed during summer compared to observations and shows too low values in October. At Pretoria, even if the $\delta^2 H_{precipitation}$ is correctly reproduced in the model, the d-excess presents differences with high values between May and September, whereas the data indicates lower values during this period. For the $^{17}O$-excess, the model-data agreement is not perfect, especially for Ankara, but the model is able to reproduce the seasonal variations as observed in the data for Reykjavik (Fig. 7d). All these model-data differences could be the result of uncertainties associated to the GNIP data and/or to biases in modelling the isotopic composition.

## 3.2 Evaluation of the main isotopic effects

### 3.2.1 Amount effect

The amount effect can be defined as a decrease of the isotopic ratio for an increase in the precipitation amount. Note that in our model the amount effect depletion is the process related to sequential precipitation removal and under-replenishment, in the form identified by Dansgaard (1964). This approach is more comparable to that of Moore et al. (2014) than the more complex approach of Risi et al. (2021). Comparing the different approaches would requires further investigation and is beyond the scope of this paper.

We investigate this effect in the model and compare it to LMDZ4 and to observations. We only extracted values in the models and for the GNIP stations that cover the tropics, from 0-20°N and 0-20°S, because this is where the amount effect is observed. For an easier comparison, we normalized the values (the raw values are presented in Appendix B for information).

The seasonal cycle in iLOVECLIM is well reproduced and in agreement with the GNIP data (especially for the precipitations between 0-20°S). In the north tropics (Fig. 8a), the isotopic ratio of the precipitation of iLOVECLIM is lower during the wet season (i.e. during the boreal summer). The opposite effect is observed in the south tropics (Fig. 8b), with high $\delta^2 H_{precipitation}$ during the austral winter, associated with a reduced amount of precipitation. So, the $\delta^2 H_{precipitation}$ decreases as precipitation intensity increases. In the model, the minimum $\delta^2 H_{precipitation}$ (maximum $\delta^2 H_{precipitation}$) is leading the minimum observed for the GNIP stations of one month (maximum observed for the GNIP stations of two months). A lag of one month is also observed between the data and LMDZ4 for the north tropics.

We further investigate this amount effect by examining the change in the $\delta^2 H_{precipitation}$ as a function of the amount of precipitation. Following Risi et al. (2008; 2010), we looked at the seasonal model variations for nine oceanic tropical GNIP stations (Apia, Barbados, Canton Island, Diego Garcia, Madang, Taguac, Truk, Wake Island and Yap). Since the resolution in iLOVECLIM is T21, the local processes may not be perfectly reproduced and complicate the comparison to local oceanic observation. Therefore, we selected for each GNIP station the pixel that was in better agreement with the precipitation and

isotopic ratio seasonal cycle data. We also do not present observational precipitation values above 350 cm y$^{-1}$ since in the model precipitations are never higher.

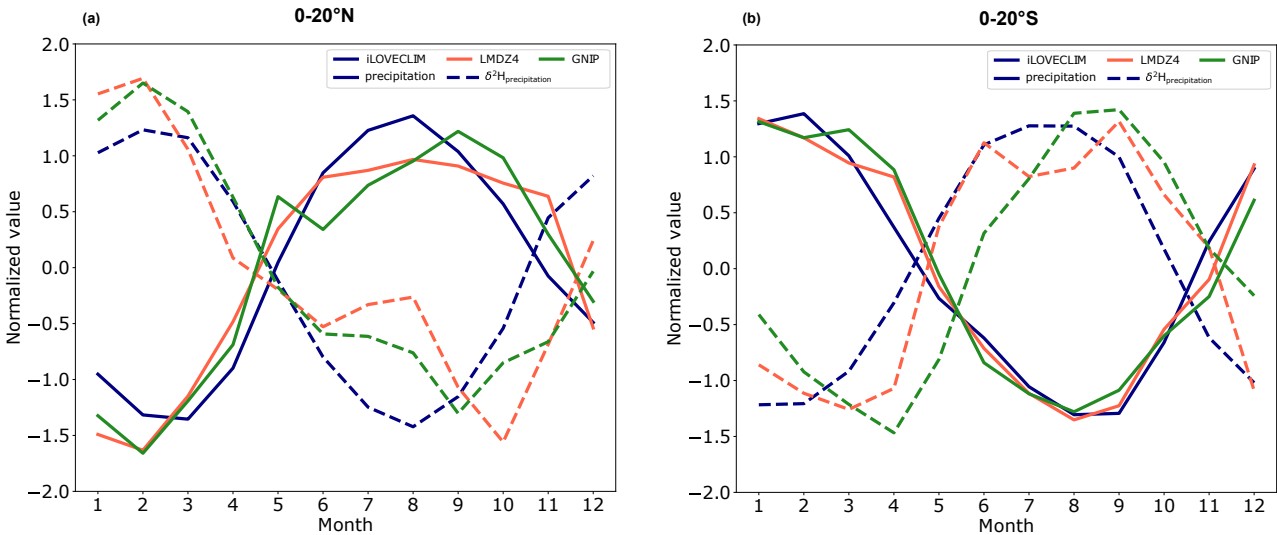

**Figure 8: Seasonal variations of the mean precipitation and δ²H$_{precipitation}$ in the tropics, from 0-20°N for (a) and from 0-20°S for (b). The values have been normalized, the solid lines represent the precipitation and the dashed lines the δ²H$_{precipitation}$. The blue curve presents the iLOVECLIM values, the red curve is for LMDZ4 and the green curve corresponds to the GNIP data.**

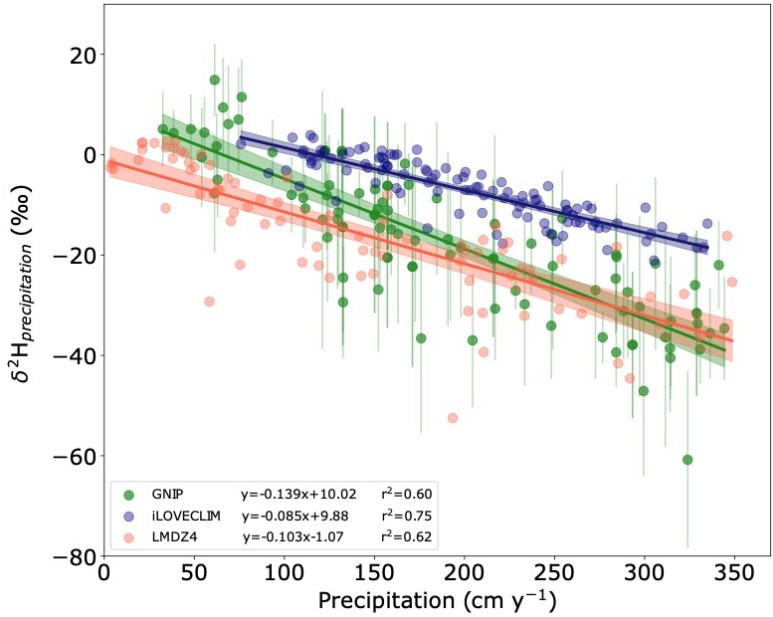

**Figure 9: Monthly δ²H$_{precipitation}$ as a function of the precipitation at the location of nine tropical oceanic GNIP stations. iLOVECLIM results in blue are compared to LMDZ4 in red and GNIP data in green. The error bars for the data are shown at 2σ.**

Figure 9 presents the relationship between the $\delta^2H_{precipitation}$ and the precipitation for the selected stations in iLOVECLIM, the observation and in LMDZ4. The isotopic ratio of precipitation is high for low precipitations and changes toward low values as precipitations increase. This amount effect is -0.085‰/cm y$^{-1}$ in iLOVECLIM, weaker than the one observed in LMDZ4 (-0.103‰/cm y$^{-1}$) and in GNIP data (-0.139‰/cm y$^{-1}$). The modelled $\delta^2H_{precipitation}$ is however higher than the observations for the same precipitation amount (especially at high precipitations). In contrast, the standard version of LMDZ4 has slightly lower $\delta^2H_{precipitation}$ at low precipitations in comparison to the observations as already noted by Risi et al. (2010).

### 3.2.2 Temperature effect

Temperature plays an important role on the hydrogen isotopic ratio of precipitation with lower values for low temperatures. We investigate in this section this relationship in iLOVECLIM and compare it to the LMDZ4 model. Since in our model the surface temperature is not a prognostic variable, we used the temperature at 650 hPa (top of the first layer) and took the equivalent temperature in LMDZ4 model at 662 hPa. An enhanced depletion of the $\delta^2H_{precipitation}$ is observed with a decrease of the temperature in both models (Fig. 10a). Differences are however noticed at low temperature (below -15°C), mainly corresponding to Antarctic values, with an isotopic ratio that is not low enough in our model. Antarctic isotopic values are indeed not computed correctly due to issues in the conservation of water in the advection scheme at very low humidity content, as already highlighted in Roche (2013). We then investigate the relationship between modelled and measured $\delta^2H_{precipitation}$, excluding Antarctic values (Fig. 10b). Most of the values are found between 0 and -60‰, with similar distribution in iLOVECLIM and LMDZ4. Differences in modelled $\delta^2H_{precipitation}$ between iLOVECLIM and LMDZ4 are enhanced for the lower values, and model-data agreement is deteriorated. As shown in Cauquoin et al. (2019b), the representation of the advection scheme in the model can impact the isotopic composition, with more enriched values when a more diffusive advection scheme is applied.

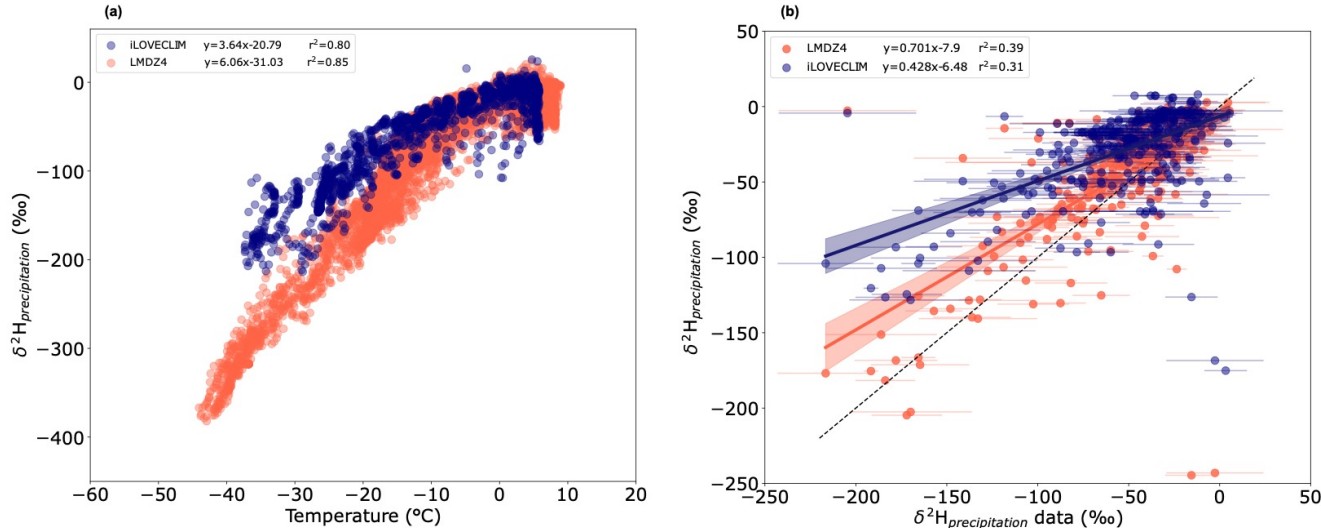

**Figure 10: (a) Annual-mean modelled $\delta^2H_{precipitation}$ as a function of the temperature for iLOVECLIM (blue) and LMDZ4 (red). (b) Annual-mean modelled $\delta^2H_{precipitation}$ for iLOVECLIM and LMDZ4 against observations (without Antarctic values). The 1:1 line is shown with the black dashed line. The errors bars associated with the data are shown at 2σ. The regression curves between model and data are presented in dark blue for iLOVECLIM and red for LMDZ4 with the confidence bands.**

### 3.2.3 Continental effect

The continental effect can be defined by a contrast in isotopic value between land and ocean, with lower values over land (Rozanski et al., 1993). To evaluate this effect in iLOVECLIM, we extracted the monthly isotopic ratio of precipitation over land and ocean separately, and focus first on the tropics between 0-20°N and 0-20°S, and second on the mid to high latitudes between 40-70°N. We also extracted values from the LMDZ4 and ECHAM5-wiso models and from the GNIP stations that have at least 3 measurements for each month. The total number of points/stations over the continents and oceans for each model (increasing with a higher resolution of the model) and observation is summarized in the Table 1. Instead of representing all data points, we decided to show the monthly mean values that correspond to the continents (America, Africa and Asia/Indonesia/Australia for the tropics; Europe, Asia and North America for the mid to high latitudes) and to the oceans (Atlantic, Pacific, Indian for the tropics; Atlantic, Pacific, Arctic for the mid to high latitudes).

The contrast in isotopic value between land and ocean, with lower values over land is well observed in the GNIP data for both tropical regions (with a median value of -23 ‰ for the continents and -9.9 ‰ for the oceans in the northern tropics, and -27.9 ‰ vs -6.1 ‰ in the southern tropics, Fig. 11a). This is due to the fact that over land, the enrichment of the low-level vapor by evaporation is weaker than over the ocean. This continental effect is observed in iLOVECLIM with a median value of -11.6 ‰ over the continents and of -4.6 ‰ over the oceans for the northern tropics, and of -17 and -3.2 ‰ over the continents and oceans respectively in the southern tropics (Fig. 11b). The difference between the land and the ocean is however less

pronounced than in the GNIP data with low values of 7 ‰ in the model compared to the 13.1 ‰ between 0-20°N for the observations (13.8 vs 21.8 ‰ between 0-20°S). This smaller depletion in the isotopic ratio over land is also observed in the LMDZ4 model. The modelled median values for LMDZ4 are similar to these obtained with iLOVECLIM, despite the difference in complexity and processes represented in the atmosphere. Among all three models and surprisingly, ECHAM5-wiso least reproduces this continental effect, despite having a better horizontal resolution.

The continental effect is well observed in the mid-high latitudes between 40-70°N in the observations with a median value of -89.8 ‰ for the continents and -51 ‰ for the oceans (Fig. 11e). iLOVECLIM, LMDZ4 and ECHAM5-wiso models reproduce this continental effect with respective median values of -52 ‰, -99.8 ‰ and -109.8 ‰ for the continents and -31.3 ‰, -43.2 ‰ and -59.5 ‰ for the oceans (Fig. 11f,g,h). The amplitude of the continental effect for these mid to high latitudes is less pronounced in iLOVECLIM than in the observations (-20.7 ‰ vs -38.9 ‰), as already observed for the tropics. The continental effect is also less pronounced at low latitudes than in mid-high latitudes in our model. In comparison, LMDZ4 and ECHAM5-wiso models have higher continental effect than observations (-56.6 ‰ and -50.3 ‰ respectively, vs -38.9 ‰).

|  | 0-20°N | | 0-20°S | | 40-70°N | |
|---|---|---|---|---|---|---|
|  | Continent | Ocean | Continent | Ocean | Continent | Ocean |
| GNIP | 13 | 9 | 21 | 7 | 107 | 4 |
| iLOVECLIM | 87 | 181 | 83 | 190 | 278 | 174 |
| LMDZ4 | 248 | 520 | 217 | 550 | 766 | 357 |
| ECHAM5-wiso | 4306 | 5454 | 1623 | 5800 | 7853 | 4178 |

**Table 1: Number of GNIP stations and points in the different models that cover land surfaces and oceans between 0-20°N, 0-20°S and 40-70°N.**

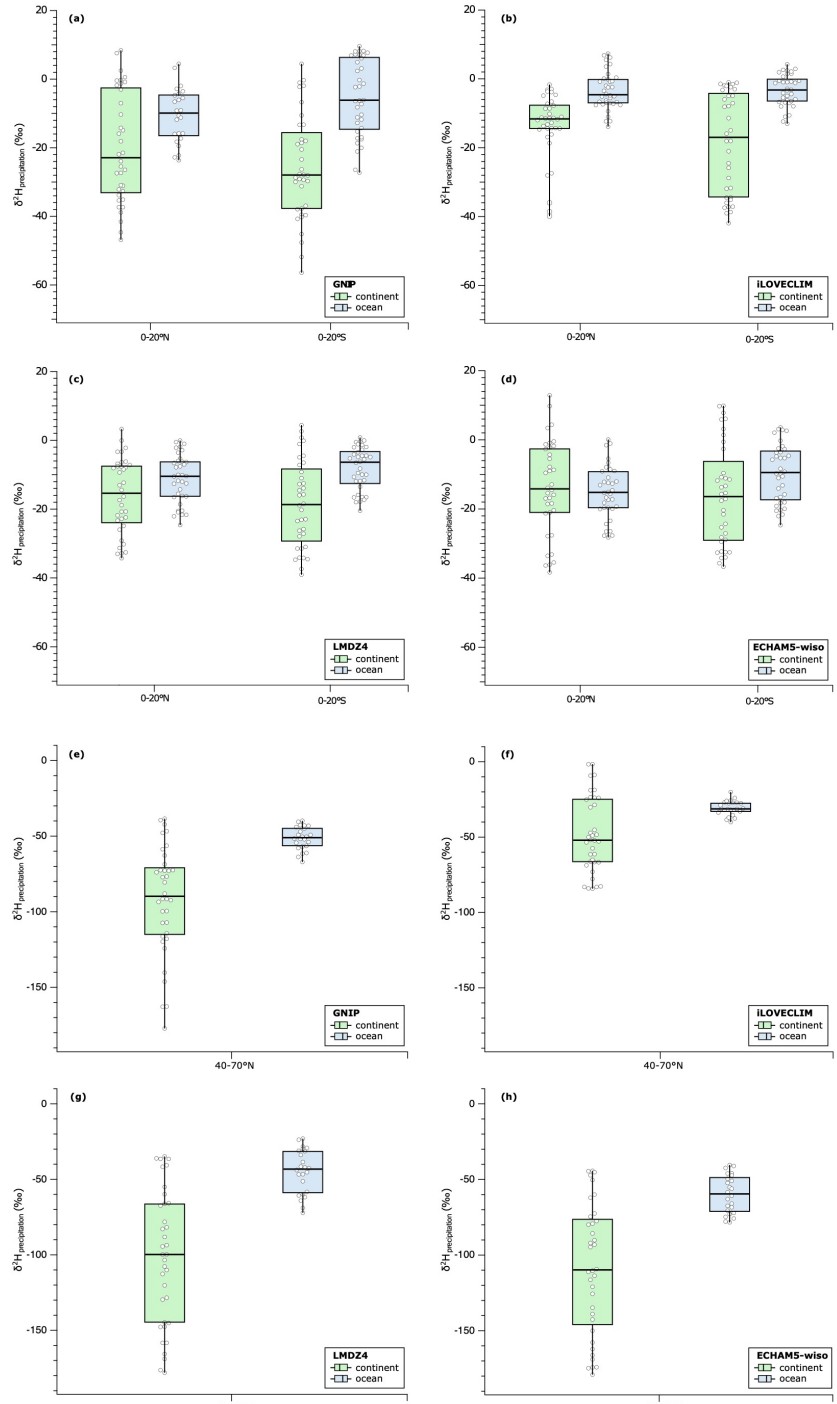

**Figure 11: Box plots of the $\delta^2H_{precipitation}$ over the continents (in green) and oceans (in blue). The panels (a) to (d) present values between 0-20°N and 0-20°S for (a) GNIP data, (b) iLOVECLIM, (c) LMDZ4 and (d) ECHAM5-wiso. The panels (e) to (h) present values between 40-70°N for (e) GNIP data, (f) iLOVECLIM, (g) LMDZ4 and (h) ECHAM5-wiso. The horizontal line in the box plots corresponds to the median value.**

### 3.3 Isotopes in ocean water

### 3.3.1 Surface seawater

The hydrogen isotopic ratio has been modelled in the oceanic component for the sea water. iLOVECLIM models annual-mean surface $\delta^2H_{seawater}$ with low negative values in the Arctic Ocean, that are too high compared to observations at high latitudes (Fig. 12a). This is clearly visible in the zonal distribution (Fig. 13a – with similar methodology than Fig. 2 to take the model outputs that co-locate with the measurements and the use a polynomial regression curve) where the $\delta^2H_{seawater}$ trend in iLOVECLIM has too high values for high latitudes compared to the observations and MPI-ESM-wiso. The $\delta^2H_{seawater}$ in the

Atlantic Ocean is well reproduced in the model with high values close to the tropic and the equator, and lower values in the northern and southern part of the ocean even if the modelled values are slightly different than the observation in the northern Atlantic (Fig. 12a). The Mediterranean Sea presents a good agreement with the observation with high $\delta^2H_{seawater}$ values. The $\delta^2H_{seawater}$ pattern in the Pacific and Austral oceans is also similar to the observations. However, the western part of the Indian Ocean and Arabian Sea presents lower values of ~10 ‰ in comparison to the GISS data (Fig. 12a). This could be explained

by a model bias toward higher precipitations and reduced salinity in this area. Both the iLOVECLIM and the MPI-ESM-wiso models reproduce the zonal distribution from 50°S to 20°N in comparison to the observations. They however present differences, with a generally lower modelled $\delta^2H_{seawater}$ value in comparison to the data, and less variability in iLOVECLIM compared to MPI-ESM-wiso (Fig. 13a).

The annual-mean surface d-excess in the different oceanic basins is also presented in Fig. 12b with the measurements for comparison. The overall pattern of d-excess is similar to the one of the $\delta^2H_{seawater}$ with high positive values in the Arctic Ocean and lower values in the Atlantic, Pacific, Indian and Austral oceans. The modelled d-excess values from -2 to 0 ‰ in the Atlantic and Pacific oceans match the observations, with a gradient from low to high values from the low to the high latitudes (Fig. 12b and Fig. 13b). The western part of the Indian Ocean and the Arabian Sea again presents different values than the

observations. The model calculates a d-excess of ~2 ‰ in the western Indian ocean whereas the data have smaller values. The modelled d-excess even goes up to 14 ‰ in the Arabian Sea, due to precipitation and humidity effect. Even if a small number of data points exist in the Polar Ocean above 60°N (only few measurements in the Atlantic sector), the model reproduces too high d-excess value in comparison to the observations, that could be explained by the absence of sea ice in this simulation. Indeed, Werner et al. (2016) shown that fractionation happen during sea ice formation, leading to depletion of the liquid surface

water isotopic composition of several permil. iLOVECLIM also does not include river discharge that are at the origin of low isotopic values and could allow for lower d-excess than in our simulation. The iLOVECLIM model presents however a closer agreement with the measurements from the mid-latitudes to the equator than the MPI-ESM-wiso model (Fig. 13b).

As for $^{17}O$-excess, modelled values are very low in the entire Arctic Ocean, Arabian Sea, Mediterranean Sea and along the

coast of east and west Africa (Fig. 12c). Apart from the northern part that has negative values similar to the Arctic Ocean, the

Atlantic Ocean presents relatively small [17]O-excess variations and matches the data with values between 0 and 50 permeg. The Pacific and Indian oceans have higher [17]O-excess values up to 200 permeg, which is higher than observations. However, considering the uncertainties associated with the model and the lack of data does not allow a good model-data evaluation for this proxy.

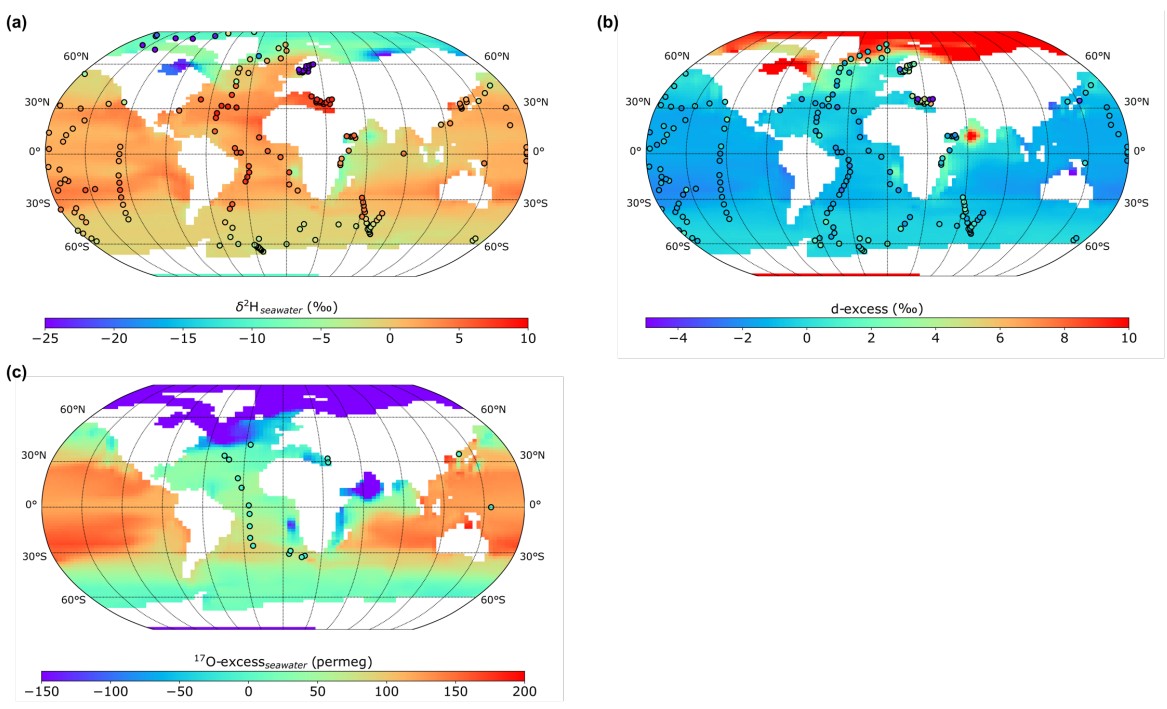


**Figure 12: Model-data comparison of the annual-mean isotopic distribution in the ocean. (a) $\delta^2$H of ocean surface water, (b) d-excess of ocean surface water and (c) [17]O-excess of ocean surface water in iLOVECLIM. The model results are compared to measurements (in circles).**


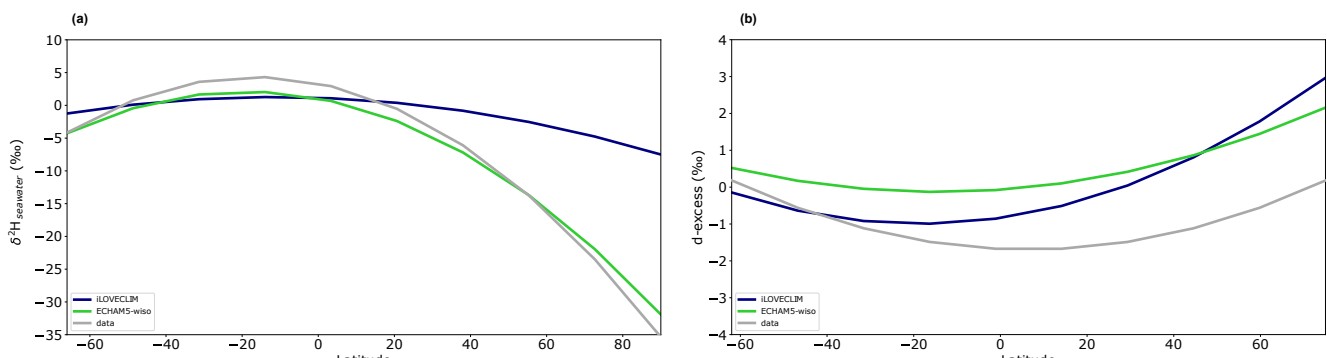

**Figure 13: Multi-model zonal comparison of (a) $\delta^2$H of ocean surface water and (b) d-excess of ocean surface water. The model results (in color) are compared to observations (in grey). The different lines are polynomial regression curves for the model results that co-locate with the observations.**

### 3.3.2 Vertical profiles

The model-data comparison of δ²H and d-excess of sea water can be realized over the entire water column with a cross section in the Atlantic Ocean. We find a general good agreement between the GISS observations and the model from the surface to the bottom with the imprint of the different water masses on the simulated δ²H (Fig. 14a). The strongest δ²H enrichment is observed in the upper Atlantic (above 700 m) between 30°S and 45°N with a maximum around 20°N with 4.2 ‰. There are however some differences in the surface water with δ²H values that are lower than the observations by several permil. Below 700 m, the North Atlantic Deep Water (NADW) have lower δ²H values, between 1.8 and up to 0 ‰ at the bottom of the ocean where they mix with the Antarctic Bottom Water (AABW) coming from the South with low values (Fig. 14a). In the Southern Ocean around 1000 m depth, the Antarctic Intermediate Water (AAIW) flow to the north with negative low δ²H values.

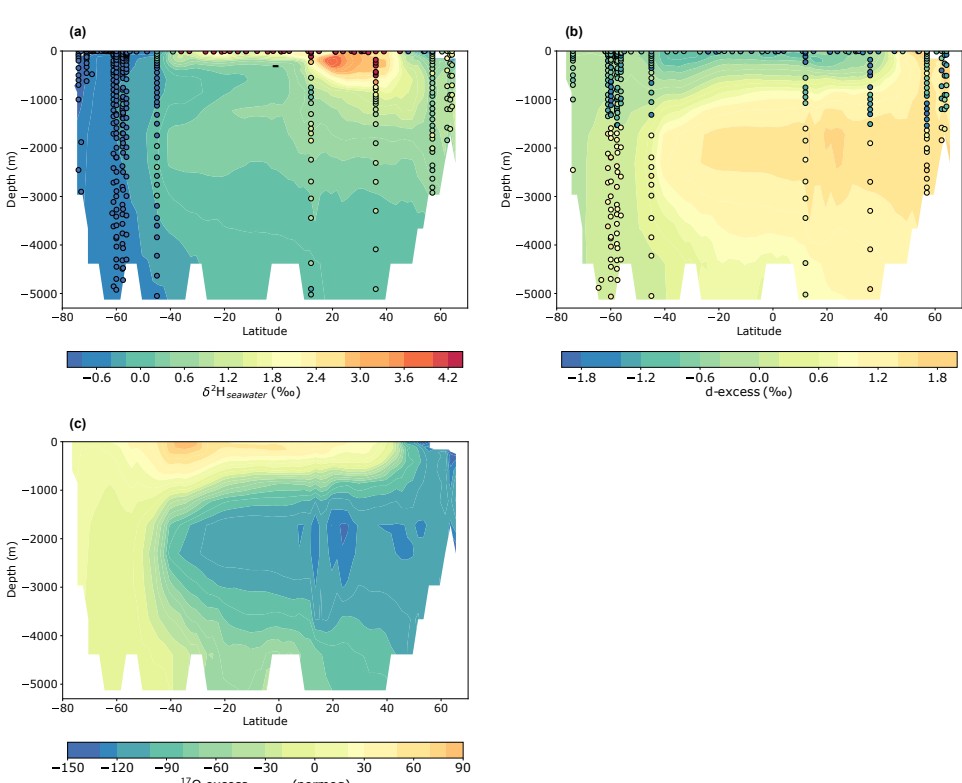

**Figure 14: Atlantic zonal mean in iLOVECLIM of (a) δ²H of seawater, (b) d-excess of seawater and (c) ¹⁷O-excess of seawater compared to observations (in circles).**

The oceanic d-excess and ¹⁷O-excess show less prominent influence of the main water masses. Above 1000 m, the d-excess goes from 40°S to 40°N with low negative values (Fig. 14b), and positive values for ¹⁷O-excess (Fig. 14c). Below 1000 m and from 40°S to the north, the NADW d-excess values are higher with a maximum of 2 ‰ around 25°N and 2000 m depth. On

the opposite, $^{17}$O-excess values are lower than in the surface, with minimum values at the same latitude and depth as the d-excess minimum. The comparison with the $\delta^2$H and d-excess observations shows that the model reproduces the low surface values and the high d-excess values below 1800 m even if the latitudinal gradient is more pronounced in the model than in the data. The depth interval from 500 to 1800 m presents a disagreement between the modelled d-excess and the observation values that are consistently lower than in the model (Fig. 14b). This is especially the case for high latitudes of the northern hemisphere where the difference between the model and the data can reach 2 to 3 ‰. Since no $^{17}$O-excess observations exist at depth, we refrain for any further evaluation of the modelled values.

**Conclusions**

In this study, we presented the implementation of the $^1$H$^2$H$^{16}$O, $^1$H$_2$$^{17}$O isotopologues in the intermediate complexity coupled climate model iLOVECLIM. Based on the existing $\delta^{18}$O water isotopic module and on this new extension, we modelled the d-excess and $^{17}$O-excess variations to have a general overview of the water isotopes. We evaluated the model isotopic ratio for preindustrial for both the atmosphere and the ocean components based on a long equilibrium simulation. For the atmospheric part, we found a good agreement between the model, the observations and several GCMs, with a reasonable simulation of the latitudinal gradient (considering the intrinsic biases of iLOVECLIM that could lead to local inconsistencies). The modelled $\delta^2$H and $\delta^{18}$O fit with the global Meteorological Water Line and the main isotopic effect, i.e. the amount effect, temperature effect and continental effect, are well reproduced in the model. The d-excess distribution for the atmosphere is also correctly modelled at global scale in comparison to the observations and several GCMs. The isotopic ratio of oxygen and hydrogen over Antarctica present however differences of several permil in comparison to the data because of the complexity of the local processes at play that are simplified in the model. At present, our models-data comparison suggests that iLOVECLIM does not correctly reproduce the $^{17}$O-excess with an excessive dispersion of the values. Modelling the $^{17}$O-excess has to be improved in the future versions of the isotope-enabled models. New measurements are also needed with a reduction of their associated uncertainties. For the ocean, we reproduced with good agreement the modelled surface $\delta^2$H and d-excess in comparison to the existing data, except for some parts of the Arctic region and local areas in the Indian Ocean. This good agreement is conserved over the entire water column in the Atlantic Ocean, with similar $\delta^2$H values and distribution between the model and the data, influenced by the main water masses.

Given the computing resources needed to run coupled climate models, applying intermediate complexity coupled climate models with water isotopes like iLOVECLIM to future long-term palaeoclimate perspectives appear very promising. Paleoclimate simulations during the Holocene, Last Glacial Maximum or transient glacial/interglacial periods are the next logical step to compare model results against past isotopic ratio records. New proxies that depend on the water isotopes can also be implemented in the model, like the leaf wax isotopic ratio, in order to quantify the influence of the respective factors (precipitation, vegetation, humidity…) that control its variations.

## Appendix A: δ¹⁷O isotopic composition

The latitudinal gradient and the global distribution for the modelled $\delta^{17}O$ is similar to the one of the $\delta^{18}O$ with low values from the equator to the poles (Fig. A1a). Similarly, the values over land are lower than over the ocean. In comparison to the available data (including new data from Terzer-Wassmuth et al., 2023), iLOVECLIM models higher values of several permil in central Europe and Canada, and lower values in Africa. Agreements are observed between the model and the data in East Asia, western Europe and North America. The discrepancies can be explained by the fact that the most of the data is punctual and reflect seasonal conditions whereas the model outputs are annual-mean $\delta^{17}O$ values.

$\delta^{17}O$ of seawater in iLOVECLIM shows values close to zero over the Atlantic, Pacific, Indian and Southern oceans which is consistent with the observations (Fig. A1b). The amplitude of variation is small and around 1 ‰. The coast of east Africa and the Arabian sea present lower values, as well as the northern part of the Atlantic Ocean and the Arctic Sea with negative values up to -4 ‰.

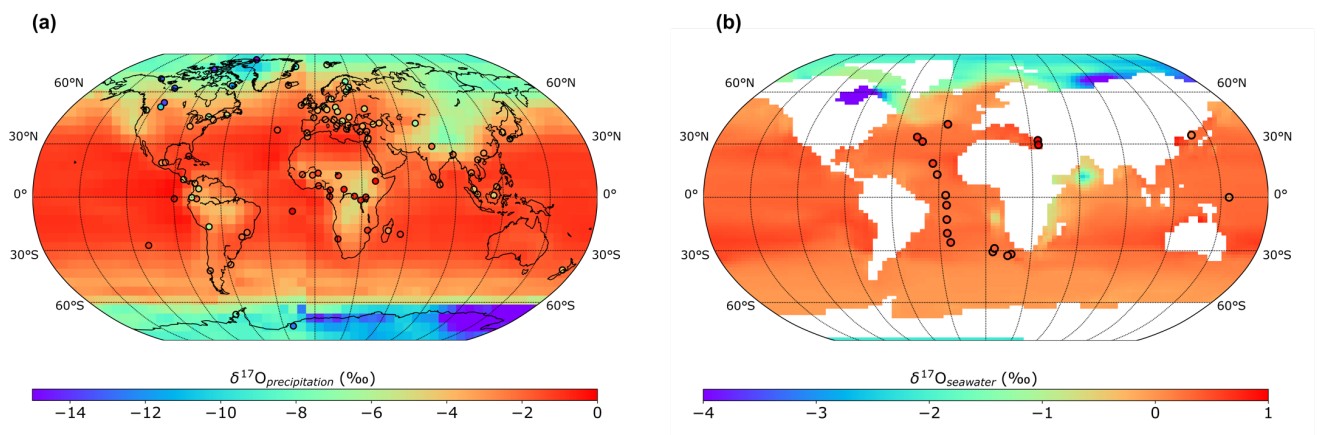

**Figure A1: Mean annual spatial distribution of the iLOVECLIM modelled (a) $\delta^{17}O_{precipitation}$ and (b) $\delta^{17}O$ of ocean surface. Model results are compared to observations (in circles).**

Figure A2 presents the relationship between modelled and measured $\delta^{17}O_{precipitation}$ (excluding values in Antarctica). Most of the values modelled in iLOVECLIM are grouped around high isotopic values, but the correlation remains low. The model results are statistically significant with a p-value of 0.007 (<0.05). In comparison to LMDZ4 that is currently the only GCM to include the $^{17}O$ (Risi et al., 2013), iLOVECLIM results are in good agreement with most of the values between 0 and -7 ‰, leading to similar linear trend between the model and the data. Towards negative values, LMDZ4 gets closer to the 1:1 line than iLOVECLIM. However, considering the large confidence intervals for both model results, the modelled $\delta^{17}O_{precipitation}$ in iLOVECLIM could be in agreement with the values obtained in LMDZ4. The differences between the model results and the

data could be related to the fact that most of the data is punctual and reflect seasonal conditions whereas the model outputs are annual-mean $\delta^{17}O$ values and to the low number of measurements to compare with.

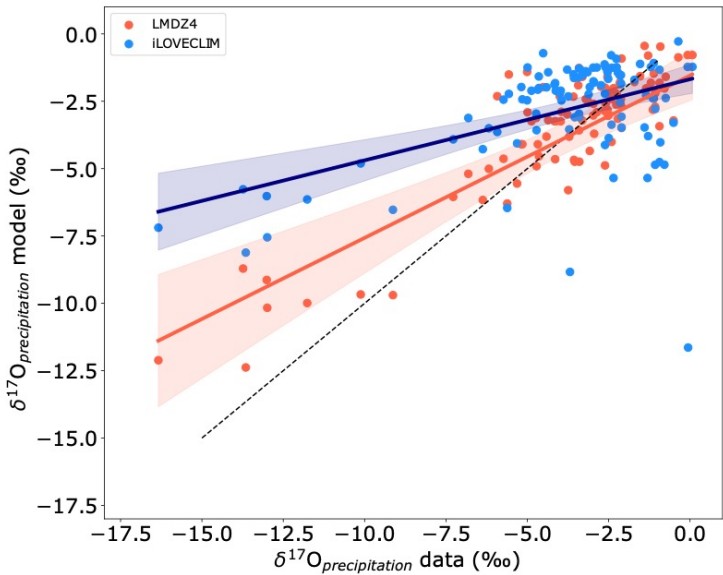

**Figure A2: Model-data relationship for the $\delta^{17}O_{precipitation}$ without Antarctic values for the iLOVECLIM (blue) and LMDZ4 (red) models. The regression curves between model and data are presented in dark blue for iLOVECLIM and red for LMDZ4 with the confidence bands. The 1:1 line is shown with the black dashed line.**

## Appendix B: Seasonal variations

We investigate the amount effect by looking at seasonal variations of the precipitation and isotopic ratio. For an easier comparison in the main text, we normalized the values because the seasonal evolution in the model is not expected to perfectly reflect the measurements. We present here in Figure B1 the raw values. The seasonal variation of the precipitation and $\delta^2H_{precipitation}$ is the same than the one presented in Section 3.2.1 with the normalized values (Fig. 8). The lead and lag of iLOVECLIM and LDMZ4 models to the data is also conserved. Differences are however observed in the amplitude, mostly for the isotopic ratio, with lower values up to 15 ‰ in summer for the north tropics between the data and the models. Same difference in absolute values between the observation and the models is observed in the south tropics.

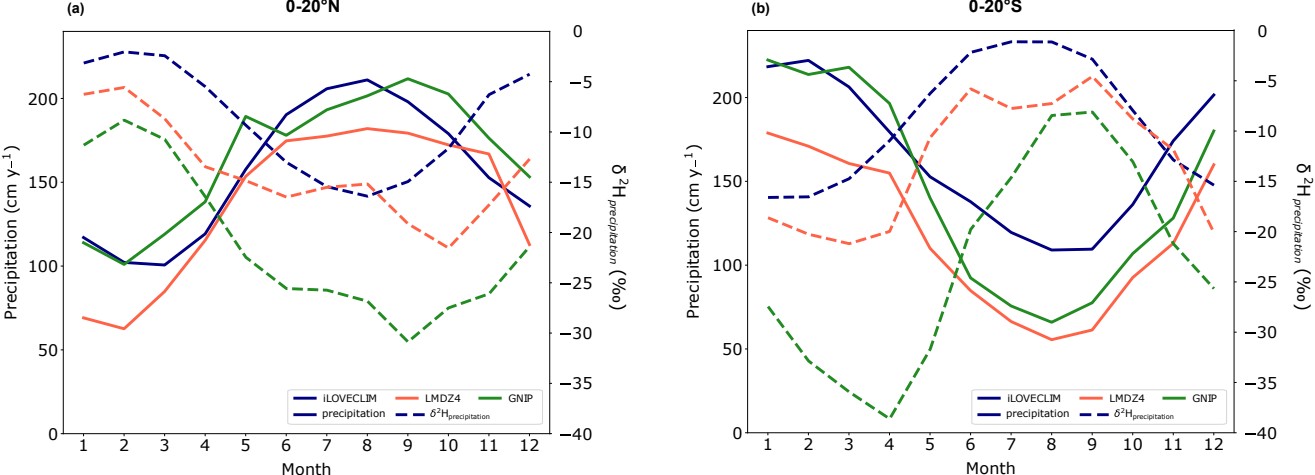

**Figure B1: Seasonal variations of the mean precipitation and δ²H_precipitation in the tropics, from 0-20°N for (a) and from 0-20°S for (b). The solid lines represent the precipitation and the dashed lines the δ²H_precipitation. The blue curve presents the iLOVECLIM raw values, the red curve is for LMDZ4 and the green curve corresponds to the GNIP data.**

*Author contributions.* TE and TC designed the study. DMR realized the model development. TE performed and analysed the simulations with inputs from TC. TE wrote the paper with contributions from all co-authors.

*Competing interest.* The authors declare that they have no conflict of interest.

*Code availability.* The iLOVECLIM source code and developments are hosted at http://forge.ipsl.jussieu.fr/ludus (IPSL, 2023) but are not publicly available due to copyright restrictions. Access can be granted on demand by request to D.M. Roche (didier.roche@lsce.ipsl.fr) to those who conduct research in collaboration with the iLOVECLIM user group.

*Financial support.* This research was supported by the ANR HYDRATE project, grant ANR-21-CE01-0001-01 of the French Agence Nationale de la Recherche.

*Acknowledgments.* T.C. is supported by CNRS-INSU.

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
