# Peer review of "Modelling water isotopologues (1H2H16O, 1H217O) in the coupled numerical climate model iLOVECLIM (version 1.1.5)"

_Geoscientific Model Development, 2023_

## Referee Comment (RC1)

**Review of Extier et al**

**July 23, 2023**

This article presents the implementation of new isotopic tracers in the iLOVECLIM intermediate complexity model. Up to now only $\delta^{18}O$ was implemented in iLOVECLIM. Now $\delta^2 H$ and $\delta^{17}O$ are also implemented. This allows to simulate some Deuterium-based paleoclimate proxies, such as the $\delta D$ of leaf waxes, and the second order isotopic parameters d-excess and $^{17}O$-excess. This article is very important because it opens the door to the use of iLOVECLIM to interpret past variations of a larger range of isotopic proxies.

The article is generally well written. I have however several major comments that I think would need to be addressed to give more confidence that iLOVECLIM is working properly and is a safe model to use for some paleoclimate applications (to be precised).

**1   Major comments**

**1.1   More quantitative and honest assessment about the skills of iLOVECLIM**

First of all, using intermediate complexity models to interpret paleoclimate proxies has advantages relative to using GCMs. I think these advantages could be better emphasized, e.g. in introduction.

These advantages come at the expense of a less realistic representation of the climate and isotopic composition. It is expected and there is no shame about it. As a reader, my main motivation for reading this article was to address the following question: For what kind of paleoclimate applications is using iLOVECLIM relevant and safe? For what aspects of isotopic variations is iLOVECLIM realistic enough? I think this article could to a better job at addressing these questions.

The article already presents comparisons between the skills of iLOVECLIM and GCMs. So all the material is already there to address these questions. It just needs to be integrated into a more quantitative comparison. For example:

- Calculate some skill metrics for iLOVECLIM and other GCMs, e.g. model-observation correlations, root-mean-square errors, for $\delta D$, d-excess, $^{17}O$-excess, possibly in different regions (e.g. entire globe or tropics). The metrics could be summarized in a table or in Taylor plot diagrams for example.

- In addition to maps, it would be very helpful to assess to what extent iLOVECLIM can simulate the main isotopic effects relative to observations: amount effect (scatter plot of $\delta D$ over tropical islands), temperature effect (scatter plot of $\delta D$ as a function of temperature), continental effect (e.g. box and whisker plots of $\delta D$ within 20°S-20°N over land and over ocean). The simulation of these effects could be compared between iLOVECLIM and other GCMs.
  I think that evaluating the isotopic effects is essential for a model that is supposed to be used for paleoclimate applications in the future, because isotope effects are an essential ingredient of paleoclimate variations in isotopes.

Based on this more quantitative comparison, clear and honest statements could be formulated about in which region, for which variables and for which isotope effects iLOVECLIM is realistic and could be used for paleoclimate applications.

**1.2   Suspected problem in the treatment of land evapo-transpiration**

- l 155-163 needs to be clarified. l 161 writes that "In the same way, evapo-transpiration occurs from the soil bucket water with fractionation": so what was equation 10 about? Evapo-transpiration represents both evaporation from soils and standing water and transpiration from plants. I cannot think of any water flux between the land and the atmosphere that is not evapo-transpiration.

- Why assuming that there is fractionation during evapo-transpiration? Evapo-transpiration is dominated by transpiration (e.g. [Jasechko et al., 2013]) which does not fractionate. Transpiration does not fractionate because no fractionation is associated with root uptake [Washburn and Smith, 1934], the water is transported by the xylem to the leaves without any fractionation, and then the water reservoir in leaves is smaller than the evaporation flux during a day. In all GCMs that are coupled to simple bucket models, evapo-transpiration is assumed not to fractionate (e.g. [Hoffmann et al., 1998, Risi et al., 2010]).

- When coupled to more sophisticated land surface models, transpiration is still assumed not to fractionate (e.g. [Haese et al., 2013, Risi et al., 2016]). The bare soil evaporation is assumed to fractionate, but never at equilibrium like equation 10. Rather, the [Craig and Gordon, 1965] equation is assumed, with specific kinetic fractionations for the soil (e.g. [Mathieu and Bariac, 1996, Barnes and Allison, 1988, Haese et al., 2013, Risi et al., 2016]).

- The fraction of bare soil evaporation (fractionating) and transpiration (non-fractionating) impacts the isotopic composition of the precipitation over land regions [Haese et al., 2013, Risi et al., 2016]. The non-fractionating transpiration is known to be essential for determining the isotopic gradients over the Amazon, Congo basin and Eastern Africa [Salati et al., 1979, Levin et al., 2009, Worden et al., 2021, Shi et al., 2022] and might also play a role in isotopic changes during past climates [Pierrehumbert, 1999]. I suspect that the big depletion bias simulate over tropical land (Fig 1a) could be partially due to the assumed fractionation during evapo-transpiration. I would recommend to re-run a new simulation without any fractionation during evapo-transpiration. This might help to improve the simulation.

**1.3 Suspected problem in the simulation of $^{17}O$-excess**

When looking at Fig 1d or Fig 6b, I'm very worried about the $^{17}O$-excess simulation. Those extreme values from -150 to 150 permeg look very strange. The spatial pattern also looks strange. What would cause such a strange pattern? The LMDZ simulation of $^{17}O$-excess, for example, was much smoother and didn't show this spatial pattern at all.

Why is $^{17}O$-excess so noisy in Antarctica and Southern Ocean? Is it a problem with large inter-annual variations and a too short simulation period? Or simply a bug? The traverse data from [Pang et al., 2015] shows much smoother variations. Given the scarcity of $^{17}O$-excess observations, the data from [Pang et al., 2015] deserves to be used and cited in this paper.

The dataset from [Uemura et al., 2010] deserves to be used and cited as well. I understand that is was in the vapor and iLOVECLIM does not allow for a model-data comparison of the vapor. Yet this dataset provides useful information: it shows that over the ocean, the $^{17}O$-excess varies very smoothly and is mainly controlled by the surface relative humidity. This observation makes the simulation by iLOVECLIM all the more suspect.

I understand that $^{17}O$-excess is very difficult to simulate in models. I don't think that a proper simulation of $^{17}O$-excess is a necessary condition to publish this paper. However, I do think that honest statements about the failure of iLOVECLIM to simulate $^{17}O$-excess, and speculation on the causes for this failure, would be very valuable.

For example, l 320-321: "could get closer": could also get further away... I don't think it is very honest to pretend that the model-data disagreement is due to uncertainties. Given the completely different ranges of $^{17}O$-excess values for observations and iLOVECLIM showed in Fig 6b, and given the smooth variations that have been reported in all $^{17}O$-excess observations so far, I think the authors can state with a high degree of confidence that iLOVECLIM fails to properly simulate $^{17}O$-excess.

**2 Minor comments**

- l 11: remove "and numerical models": we don't need isotopes to infer hydrological changes in numerical models, these can be directly diagnosed by outputting all necessary variables.

- l 24: compare -> compared

- l 24: And? The reader here expects a sentence assessing the skill of iLOVECLIM for $^{17}O$-excess. This is a key aspect of the paper and it needs to be in the abstract.

- l 53: "new method ... 2006... 2008." I don't think methods published more than 15 years ago can still be called "new".

Generally, I found that many references in this paper were quite old, maybe some bibliographic update could be useful.

- l 90: "500 and 200hPa ... dry layers correspond to the stratosphere": Does it make any sense that the stratosphere is so low in altitude? Don't these levels simply represent the free troposphere?

- l 110-115: Equation 4 is simply the [Craig and Gordon, 1965] equation. This is the equation used in all isotope-enabled model, including all GCMs. I don't know why the authors introduce it in such a complicated way, and why Cappa et al 2003 and Roche 2013 need to be cited for this. Rather, [Craig and Gordon, 1965] should be cited.
"The evaporation term...": Really? If you write the bulk evaporation equation for humidity, and the same for isotopes, you take the ratio, and you find the [Craig and Gordon, 1965]. So I don't understand the problem.

- $R_a$, $h_a$: what do they represent? Does it represent the isotopic composition of the 800hPa layer? Physically, does it represent the "free atmosphere", or the boundary layer? If this really represents the "free atmosphere", does this lead to a systematic bias, with too depleted $R_a$? Is there a correction to account for this?
"free atmosphere": should rather be "free troposphere"?

- l 128: "tropopause, mid-troposphere": what do these levels correspond to? Do these correspond to 200hPa and 500hPa respectively? It would be clearer to refer to the levels in hPa rather.

- l 150: is there any representation of evaporation of rain as it falls? Rain evaporation is known to be essential for simulating the amount effect, e.g. [Field et al., 2010, Risi et al., 2021]. If there is no rain evaporation, does it mean that the iLOVECLIM cannot represent the amount effect? Does it mean that any use of iLOVECLIM to interpret paleoclimate proxies in tropical regions is problematic?
It would be very helpful to show to what extent iLOVECLIM is able to simulate the amount effect, see major comment 1. From Fig 1 it looks like it is not, but it's hard to see on a map.

- l 174: what is the time step of the model?

- l 205: "Risi et al 2012" can be used as a reference for SWING2, but for LMDZ4, replace by [Risi et al., 2010].

- l 205: are all these simulations part of the SWING2 database? If so, write it.

- l 210: "better reproduce isotopic change above 80°N than in the other models": I cannot see this in Fig 2. There isn't any observation above 80°N in this Fig.

- Fig 1: I don't think the map for $\delta^{17}O$ is useful, since it shows exactly the same as for $\delta^2 H$. The added value of $\delta^{17}O$ relative to other isotopes is already well summarized by $^{17}O$-excess in Fig 1d.
In contrast, I think that it would be worth to show the model-observation comparison for temperature and precipitation, because these variables can help interpret some of the model biases for isotopes.

- Fig 2: were the model outputs co-located with the measurements? For a more rigorous comparison, it might be useful to do so.

- Fig 2: it would be useful to have the same for d-excess and $^{17}O$-excess. More generally, it looks like there is a new figure style for each isotopic variable. It would help the reader to have more coherent figures between the different variables. e.g. zonal mean for $\delta^2 H$, d-excess and $^{17}O$-excess, same style of model-obs scatter plot for $\delta^2 H$, d-excess and $^{17}O$-excess, etc...

- If there are too many figures, I think Fig 3 is not so useful. The MWL is not a stringent test on the simulations.

- l 285-290, 301-310: maybe these paragraphs could be summarized by just noticing that the spatial pattern of $\delta^{17}O$ looks almost exactly the same as $\delta^{18}O$? The $^{17}O$-excess parameter is what bears the added value.

- l 311: "proxy" -> "variable". For present day, $\delta^{17}O$ is directly measured.

- Fig 9: same for d-excess?

- l 424: "relatively similar close to zero values" -> values close to 0‰.
  Same problem l200

- l 443: remove "a better agreement... at least", because only the second part of the sentence is correct.

- Fig A1: I think this figure should replace Fig 4 in the text, and the appendix text can be merged in the main text. Everything that could be seen in Fig 4 can be seen in A1.

- Please check the reference list. Some articles cited in the text are missing, e.g. Werner et al 2011.

**References**

[Barnes and Allison, 1988] Barnes, C. and Allison, G. (1988). Tracing of water movement in the unsaturated zone using stable isotopes of hydrogen and oxygen. *J. Hydrol*, 100:143–176.

[Craig and Gordon, 1965] Craig, H. and Gordon, L. I. (1965). Deuterium and oxygen-18 variations in the ocean and marine atmosphere. *Stable Isotope in Oceanographic Studies and Paleotemperatures*, Laboratorio di Geologia Nucleate, Pisa, Italy:9–130.

[Field et al., 2010] Field, R. D., Jones, D. B. A., and Brown, D. P. (2010). The effects of post-condensation exchange on the isotopic composition of water in the atmosphere. *J. Geophy. Res.*, 115, D24305:doi:10.1029/2010JD014334.

[Haese et al., 2013] Haese, B., Werner, M., and Lohmann, G. (2013). Stable water isotopes in the coupled atmosphere-land surface model ECHAM5-JSBACH. *Geoscientific Model Development*, 6:1463–1480, doi: 10.5194/gmd–6–1463–2013.

[Hoffmann et al., 1998] Hoffmann, G., Werner, M., and Heimann, M. (1998). Water isotope module of the ECHAM atmospheric general circulation model: A study on timescales from days to several years. *J. Geophys. Res.*, 103:16871–16896.

[Jasechko et al., 2013] Jasechko, S., Sharp, W. D., Sharp, J. J., Birks, S. J., Yi, Y., and Fawcett, P. J. (2013). Terrestrial water fluxes dominated by transpiration. *Nature*, 496:347–350, doi:10.1038/nature11983.

[Levin et al., 2009] Levin, N. E., Zipser, E. J., , and Cerling, T. E. (2009). Isotopic composition of waters from Ethiopia and Kenya:Insights into moisture sources for eastern Africa. *J. Geophys. Res.*, 114:D23306, doi:10.1029/2009JD012166.

[Mathieu and Bariac, 1996] Mathieu, R. and Bariac, T. (1996). A numerical model for the simulation of stable isotope profiles in drying soils. *J. Geophys. Res.*, 101 (D7):12685–12696.

[Pang et al., 2015] Pang, H., Hou, S., Landais, A., Masson-Delmotte, V., Prie, F., Steen-Larsen, H. C., Risi, C., Li, Y., Jouzel, J., Wang, Y., et al. (2015). Spatial distribution of 17o-excess in surface snow along a traverse from zhongshan station to dome a, east antarctica. *Earth and Planetary Science Letters*, 414:126–133.

[Pierrehumbert, 1999] Pierrehumbert, R. T. (1999). Huascaran delta18O as an indicator of tropical climate during the Last Glacial Maximum. *Geophys. Res. Lett.*, 26:1345–1348.

[Risi et al., 2016] Risi, C., Bony, S., Ogée, J., Bariac, T., Raz-Yaseed, N., Wingate, L., Welker, J., Knohl, A., Kurz-Besson, C., Leclerc, M., Zhang, G., N, B., Santrucek, J., Hronkova, M., David, T., Peylin, P., and Guglielmo, F. (2016). The water isotopic version of the land-surface model ORCHIDEE: implementation, evaluation, sensitivity to hydrological parameters. *Hydrology: Current Research*, 7:DOI: 10.4172/2157–7587.1000258.

[Risi et al., 2010] Risi, C., Bony, S., Vimeux, F., and Jouzel, J. (2010). Water stable isotopes in the LMDZ4 General Circulation Model: model evaluation for present day and past climates and applications to climatic interpretation of tropical isotopic records. *J. Geophys. Res.*, 115, D12118:doi:10.1029/2009JD013255.

[Risi et al., 2021] Risi, C., Muller, C., and Blossey, P. (2021). Rain evaporation, snow melt, and entrainment at the heart of water vapor isotopic variations in the tropical troposphere, according to large-eddy simulations and a two-column model. *J. Adv. Model. Earth Sci.*, 13(4):e2020MS002381, DOI: https://doi.org/10.1029/2020MS002381.

[Salati et al., 1979] Salati, E., Dall'Olio, A., Matsui, E., and Gat, J. (1979). Recycling of water in the Amazon basin: An isotopic study. *Water Resources Research*, 15:1250–1258.

[Shi et al., 2022] Shi, M., Worden, J. R., Bailey, A., Noone, D., Risi, C., Fu, R., Worden, S., Herman, R., Payne, V., Pagano, T., et al. (2022). Amazonian terrestrial water balance inferred from satellite-observed water vapor isotopes. *Nature communications*, 13(1):1–10.

[Uemura et al., 2010] Uemura, R., Barkan, E., Abe, O., and Luz, B. (2010). Triple isotope composition of oxygen in atmospheric water vapor. *Geophy. Res. Lett.*, 37:L04402, doi:10.1029/2009GL041960.

[Washburn and Smith, 1934] Washburn, E. and Smith, E. (1934). The isotopie fractionation of water by physiological processes. *Science*, 79:188–189.

[Worden et al., 2021] Worden, S., Fu, R., Chakraborty, S., Liu, J., and Worden, J. (2021). Where does moisture come from over the congo basin? *Journal of Geophysical Research: Biogeosciences*, 126(8):e2020JG006024.

---

## Author Comment (AC1)

**Response to Reviewer 1**

We thank Reviewer 1 for the detailed review and suggestions which helped to improve the manuscript. We are providing our answers (in blue) to the comments and will revise the manuscript accordingly.

- **Major comments**

**1.1 More quantitative and honest assessment about the skills of iLOVECLIM**

First of all, using intermediate complexity models to interpret paleoclimate proxies has advantages relative to using GCMs. I think these advantages could be better emphasized, e.g. in introduction. These advantages come at the expense of a less realistic representation of the climate and isotopic composition. It is expected and there is no shame about it. As a reader, my main motivation for reading this article was to address the following question: For what kind of paleoclimate applications is using iLOVECLIM relevant and safe? For what aspects of isotopic variations is iLOVECLIM realistic enough? I think this article could to a better job at addressing these questions.

Thank you for this comment. We added a new paragraph in the introduction to synthetize previous work the work on isotope modelling with the climate models and to detail the applications with iLOVECLIM: "*Since the initial works of Joussaume et al. (1984) and Jouzel et al. (1987), much progress has been done in atmospheric general circulation models (AGCMs) (e.g. Hoffmann et al., 1998; Noone and Simmonds, 2002; Mathieu et al., 2002, Risi et al., 2010; Werner et al., 2011) that can simulate accurately the $\delta^{18}O$ of precipitation. The subsequent development of water isotopes modules in oceanic general circulation models (OGCMs) (Schmidt, 1998; Delaygue et al., 2000; Xu et al., 2012) opens the possibility for coupled simulations of present and past climates, conserving water isotopes through the hydrosphere (Schmidt et al., 2007; Zhou et al., 2008; Tindall et al., 2009; Werner et al., 2016; Cauquoin et al., 2019). In general, General Circulation Models (GCMs) have been used exclusively to simulate separately water isotopes in the atmospheric and oceanic components. Given the computing resources needed to run coupled climate models, applying intermediate complexity coupled climate models with water isotopes like iLOVECLIM to long-term palaeoclimate perspectives still appears quite suitable (e.g Caley et al., 2014). It could allow to improve our understanding of the relationship between water isotopologues, second-order parameter (like d-excess) and climate over a broad range of simulated climate changes*".

In this paper we only investigated the capacity of our model to reproduce the hydrogen isotopic composition, d-excess and $^{17}O$-excess for present day, as it is a development paper. The simulation of the isotopic composition under another past climate period is not within the scope of this paper. But this opens new possibilities to perform long-term transient simulations with a model equipped with the isotopes since iLOVECLIM has the possibility to run simulations over several thousands of years within several weeks/months. For example, paper like Caley et al. (2014) already investigated past changes in the modelled oxygen isotopic composition during a glacial-interglacial cycle. We added a sentence in the manuscript to emphasize this aspect: "*Given the computing resources needed to run coupled climate models, applying intermediate complexity coupled climate models with water isotopes such iLOVECLIM to future long-term palaeoclimate perspectives appear very promising. Paleoclimate simulations during the*

*Holocene, Last Glacial Maximum or transient glacial/interglacial periods are the next logical step to compare model results against past isotopic composition records*".

Calculate some skill metrics for iLOVECLIM and other GCMs, e.g. model-observation correlations, root- mean-square errors, for δD, d-excess, $^{17}$O-excess, possibly in different regions (e.g. entire globe or tropics). The metrics could be summarized in a table or in Taylor plot diagrams for example.

We added in the revised version of the manuscript a Taylor diagram to summarize some metrics (correlation coefficient R, standard deviation SD and root mean square error RMSE) between the models equipped with the isotopic composition and the observations. This new figure compiles the correlation between several water isotopes-enabled models and the GNIP observations for the $\delta^2$H (Fig. 1a), d-excess (Fig. 1b), $^{17}$O-excess (Fig. 1c) respectively. We do not include the Antarctic values since we cannot properly reproduce the isotopic variations at these latitudes as a consequence of the non-conservative behaviour of the advection scheme at very low moisture content (as already explained). We added the description of these metrics in the different isotopic sections of the revised manuscript relative to the $\delta^2$H (Section 3.1.1), d-excess (Section 3.1.2) and $^{17}$O-excess (Section 3.1.3).

We observe for the $\delta^2$H$_{precipitation}$ that ECHAM5-wiso is the model that has the best correlation coefficient with the observation (R=0.64 vs R=0.56 for iLOVECLIM). The different GCMs have close correlation coefficient (between 0.59 and 0.64), standard deviation (between 40.21 and 46.43) and RMSE (between 34.94 and 39.82). The iLOVECLIM model presents a lower standard deviation (SD=29.93) and RMSE than the other models (Fig. 1a). However, considering the close metrics between all models, iLOVECLIM presents the advantage to run faster than other GCMs and is perfectly justified for the use of long-term global climate simulation.

[Figure]

*Figure 1: Taylor diagram representing (a) δ$^2$Hprecipitation, (b) d-excess and (c) $^{17}$O-excess values for different climate models (iLOVECLIM, LMDZ4, ECHAM5-wiso, CAM, GISS and MIROC) without Antarctic values. The simulated values are plotted against the observations. The dotted curved line indicates the reference line (standard deviation of the observation) and the bold grey contours represent RMSE values.*

For the d-excess and in comparison to the measurements for the atmosphere, iLOVECLIM has a correlation coefficient that is in the range of others models (0.34 to 0.52), but has a higher SD compared to the observations and other GCMs. The CAM model has the best correlation coefficient with the observations whereas LMDZ4 has the closest standard deviation relative to the observations (Fig. 1b). Within all models, MIROC is the one with the

lowest SD and RMSE. However, considering the general low correlation coefficient for all models, they all do not perfectly reproduce the d-excess variations as observed in the data. iLOVECLIM however presents the advantage to run faster than the other GCMs and could be used to investigate past changes in d-excess in global transient simulations.

We observe for the [17]O-excess a low correlation coefficient for iLOVECLIM and a low negative correlation coefficient for LMDZ4 with respect to observations. The standard deviation and root mean square error is better for LMDZ4 than for iLOVECLIM (Fig. 1c), suggesting that our model does not correctly reproduce the [17]O-excess and has a too important dispersion of the values, even if the trend is correct.

In addition to maps, it would be very helpful to assess to what extent iLOVECLIM can simulate the main isotopic effects relative to observations: amount effect (scatter plot of δD over tropical islands), temperature effect (scatter plot of δD as a function of temperature), continental effect (e.g. box and whisker plots of δD within 20°S-20°N over land and over ocean). The simulation of these effects could be compared between iLOVECLIM and other GCMs.

I think that evaluating the isotopic effects is essential for a model that is supposed to be used for paleoclimate applications in the future, because isotope effects are an essential ingredient of paleoclimate variations in isotopes.

Thank you for these suggestions. We added several new figures to respectively represent the amount effect, temperature effect and continental effect in the model. We also compared our results against the GNIP data and the LMDZ4 model outputs. The following is presented in the revised manuscript in a new Section 3.2 Evaluation of the main isotopic effects.

1. Amount effect

To investigate the amount effect, we first took the monthly outputs of the precipitation and $\delta^2H$ of the precipitation from the iLOVECLIM model, from LMDZ4 (Risi et al., 2010) and from the GNIP data (IAEA, 2023). We only extracted values in the models and for the GNIP stations that cover the tropics, from 0-20°N and from 0-20°S, to see if a change in precipitation intensity would lead to a change in the hydrogen isotopic composition of the precipitation. For an easier comparison, we normalized the values (we subtracted the annual mean and divided by the standard deviation).

[revised manuscript text omitted]

**1.2 Suspected problem in the treatment of land evapo-transpiration**

l 155-163 needs to be clarified. l 161 writes that "In the same way, evapo-transpiration occurs from the soil bucket water with fractionation": so what was equation 10 about? Evapo-transpiration represents both evaporation from soils and standing water and transpiration from plants. I cannot think of any water flux between the land and the atmosphere that is not evapo-transpiration.

Why assuming that there is fractionation during evapo-transpiration? Evapo-transpiration is dominated by transpiration (e.g. [Jasechko et al., 2013]) which does not fractionate. Transpiration does not fractionate because no fractionation is associated with root uptake [Washburn and Smith, 1934], the water is transported by the xylem to the leaves without any fractionation, and then the water reservoir in leaves is smaller than the evaporation flux during a day. In all GCMs that are coupled to simple bucket models, evapo-transpiration is assumed not to fractionate (e.g. [Hoffmann et al., 1998, Risi et al., 2010]).

When coupled to more sophisticated land surface models, transpiration is still assumed not to fraction- ate (e.g. [Haese et al., 2013, Risi et al., 2016]). The bare soil evaporation is assumed to fractionate, but never at equilibrium like equation 10. Rather, the [Craig and Gordon, 1965] equation is assumed, with specific kinetic fractionations for the soil (e.g. [Mathieu and Bariac, 1996, Barnes and Allison, 1988, Haese et al., 2013, Risi et al., 2016]).

The fraction of bare soil evaporation (fractionating) and transpiration (non-fractionating) impacts the isotopic composition of the precipitation over land regions [Haese et al., 2013, Risi et al., 2016]. The non-fractionating transpiration is known to be essential for determining the isotopic gradients over the Amazon, Congo basin and Eastern Africa [Salati et al., 1979, Levin et al., 2009, Worden et al., 2021, Shi et al., 2022] and might also play a role in isotopic changes during past climates [Pierrehumbert, 1999]. I suspect that the big depletion bias simulate over tropical land (Fig 1a) could be partially due to the assumed fractionation during evapo-transpiration. I would recommend to re-run a new simulation without any fractionation during evapo-transpiration. This might help to improve the simulation.

Thanks for the remark and very extended discussion of these processes. The text that is referred to was built from Roche (2013). A careful examination of the model code as used in the simulation presented in this study assumes no fractionation during all land-related evaporation processes, contrary to what was stated in the previous version of the manuscript. We removed Equation 10 and corrected the text accordingly: "*If re-evaporation occurs on land, it is assumed to be at equilibrium (without fractionation)*".

**1.3 Suspected problem in the simulation of $^{17}$O-excess**

When looking at Fig 1d or Fig 6b, I'm very worried about the $^{17}$O-excess simulation. Those extreme values from -150 to 150 permeg look very strange. The spatial pattern also looks strange. What would cause such a strange pattern? The LMDZ simulation of $^{17}$O-excess, for example, was much smoother and didn't show this spatial pattern at all. Why is $^{17}$O-excess so noisy in Antarctica and Southern Ocean? Is it a problem with large inter-annual variations and a too short simulation period? Or simply a bug?

You are right, the simulated $^{17}$O-excess in iLOVECLIM is not correctly reproduced and presents a too important dispersion of the values. This is shown when comparing with the measurements (in the Taylor diagram or in the zonal distribution plot). This is not a problem of too short simulation because the model has run 5,000 year and is at equilibrium with the climate. We instead suggest that the $^{17}$O-excess is not properly reproduced mainly because of the complex processes involved in the $^{17}$O-excess isotopic value, and because of difficulties in modelling the isotopic composition for area with very low humidity content (especially for Antarctic values).

In comparison, LMDZ4 shows indeed much smoother variations and less dispersion than iLOVECLIM but presents a general trend that is the opposite to the one observed in the measurements (Figure 6), suggesting that both models do not perfectly reproduce the $^{17}$O-excess, but probably for different reason.

We clearly state in the revised manuscript that iLOVECLIM does not correctly reproduce the $^{17}$O-excess.

The traverse data from [Pang et al., 2015] shows much smoother variations. Given the scarcity of $^{17}$O-excess observations, the data from [Pang et al., 2015] deserves to be used and cited in this paper. The dataset from [Uemura et al., 2010] deserves to be used and cited as well. I understand that is was in the vapor and iLOVECLIM does not allow for a model-data comparison of the vapor. Yet this dataset provides useful information: it shows that over the ocean, the $^{17}$O-excess varies very smoothly and is mainly controlled by the surface relative humidity. This observation makes the simulation by iLOVECLIM all the more suspect.

Thank you for pointing us these datasets. We added the data of Uemura et al. (2010) and Pang et al. (2015) to the $^{17}$O-excess figures in the revised manuscript for a better model-data comparison.

I understand that $^{17}$O-excess is very difficult to simulate in models. I don't think that a proper simulation of $^{17}$O-excess is a necessary condition to publish this paper. However, I do think that honest statements about the failure of iLOVECLIM to simulate $^{17}$O-excess, and speculation on the causes for this failure, would be very valuable. For example, l 320-321: "could get closer": could also get further away... I don't think it is very honest to pretend that the model-data disagreement is due to uncertainties. Given the completely different ranges of $^{17}$O-excess values for observations and iLOVECLIM showed in Fig 6b, and given the smooth variations that have been reported in all $^{17}$O-excess observations so far, I think the authors can state with a high degree of confidence that iLOVECLIM fails to properly simulate $^{17}$O-excess.

Based on the new figures to evaluate the model metrics like model-observation correlation or RMSE, we now properly state in the revised manuscript that the $^{17}$O-excess is not correctly reproduce in iLOVECLIM. This is based on low correlation coefficient between iLOVECLIM and the observations, on the higher dispersion of the data, standard deviation and root mean square errors than LMDZ4.

- **Line by line comments**

l 11: remove "and numerical models": we don't need isotopes to infer hydrological changes in numerical models, these can be directly diagnosed by outputting all necessary variables.

You are correct. We removed it from the text.

l 24: compare -> compared

Done.

l 24: And? The reader here expects a sentence assessing the skill of iLOVECLIM for 17O-excess. This is a key aspect of the paper and it needs to be in the abstract.

We added the following sentence in the abstract to highlight the capacity of iLOVECLIM to model the $^{17}$O-excess: "*The modelled $^{17}$O-excess presents a too important dispersion of the values in comparison to the observations and is not correctly reproduced in the model mainly because of the complex processes involved in the $^{17}$O-excess isotopic value.*".

l 53: "new method ... 2006... 2008." I don't think methods published more than 15 years ago can still be called "new".

We replaced "new method" by "another method".

l 90: "500 and 200hPa ... dry layers correspond to the stratosphere": Does it make any sense that the stratosphere is so low in altitude? Don't these levels simply represent the free troposphere?

This was indeed incorrectly formulated. The text has been modified and now reads: "*It is subdivided in three vertical layers at 800, 500 and 200 hPa with the humidity contained only in the first layer and representative of the total humidity content of the atmosphere*".

l 110-115: Equation 4 is simply the [Craig and Gordon, 1965] equation. This is the equation used in all isotope-enabled model, including all GCMs. I don't know why the authors introduce it in such a complicated way, and why Cappa et al 2003 and Roche 2013 need to be cited for this. Rather, [Craig and Gordon, 1965] should be cited. "The evaporation term...": Really? If you write the bulk evaporation equation for humidity, and the same for isotopes, you take the ratio, and you find the [Craig and Gordon, 1965]. So I don't understand the problem.

In Roche (2013) the derivation of the equation for the isotopic evaporation was based on the Cappa et al. (2003) model which is fully-derived within the text of the reference (see their Equation A9). We acknowledge that the approach of Cappa et al. (2003) is very similar to the Craig and Gordon (1965) approach and thus leads to very similar formula. The Craig and Gordon (1965) approach in itself is a family of models; we have thus modified the text accordingly which now reads: "*The solution adopted by Roche (2013) is to compute the water isotopic ratio in the evaporation using a Craig and Gordon (1965) type-model in the formulation adapted by Cappa et al. (2003)*". However, as highlighted in Roche (2013), the ECBilt model does not prognostically simulate the variables we need to be consistent with the Cappa et al. (2003) formulation, hence the use of the apparent humidity $h_a^*$ (see next question).

$R_a$, $h_a$: what do they represent? Does it represent the isotopic composition of the 800hPa layer? Physically, does it represent the "free atmosphere", or the boundary layer? If this really represents the "free atmosphere", does this lead to a systematic bias, with too depleted $R_a$? Is there a correction to account for this? "free atmosphere": should rather be "free troposphere"?

As described in Section 2.1, the model is T21L3 with humidity only in the first layer. Therefore, any variable relating to humidity in the atmosphere (such as relative humidity, water content, isotopic composition of the vapor) represents the content of the first layer that is the whole atmospheric content (see Figure 1 of Roche, 2013).

l 128: "tropopause, mid-troposphere": what do these levels correspond to? Do these correspond to 200hPa and 500hPa respectively? It would be clearer to refer to the levels in hPa rather.

As mentioned before, ECBilt is a T21L3 model with layers at 800, 500 and 200 hPa. There is indeed no assumption of the location of the tropopause. This was ill-formulated in the previous version of the manuscript which now reads: "*The precipitations (convective and large scale) and snow are in equilibrium with isotopic values at 650, 800 hPa and 650 hPa respectively*".

l 150: is there any representation of evaporation of rain as it falls? Rain evaporation is known to be essential for simulating the amount effect, e.g. [Field et al., 2010, Risi et al., 2021]. If there is no rain evaporation, does it mean that the iLOVECLIM cannot represent the amount

effect? Does it mean that any use of iLOVECLIM to interpret paleoclimate proxies in tropical regions is problematic? It would be very helpful to show to what extent iLOVECLIM is able to simulate the amount effect, see major comment 1. From Fig 1 it looks like it is not, but it's hard to see on a map.

ECBilt has no representation of re-evaporation of rain as it falls (Opsteegh et al., 1998): this is now explicitly mentioned in Section 2.1: "*Water fluxes crossing the limit between the humid and dry layers are rained out instantly as convective rain*". Regarding the amount effect, see the answer above in major comment 1.1.

l 174: what is the time step of the model?

The atmospheric module has a timestep of 6 hours and the oceanic module has a daily timestep. We added these timesteps in the respective Sections 2.1 and 2.2 for the atmosphere and the ocean.

l 205: "Risi et al 2012" can be used as a reference for SWING2, but for LMDZ4, replace by [Risi et al., 2010].

Done.

l 205: are all these simulations part of the SWING2 database? If so, write it.

We added a paragraph in the Section 2.4 Observational data and water isotopes-enabled GCMs, to list the model results used in the paper to compare with. We specify that most of them comes from the SWING2 database. The manuscript now reads: "*To evaluate our model results against water isotopes-enabled GCMs, we used several model outputs: ECHAM5-wiso (Steiger et al., 2018), GISS (Schmidt et al., 2007), LMDZ4 (Risi et al., 2010, Risi et al., 2013), MIROC (Kurita et al., 2011), CAM (Lee et al., 2007) and MPI-ESM-wiso (Cauquoin et al., 2020). The GISS, LMDZ4, MIROC and CAM data are from the Stable Water Isotope Intercomparison Group, Phase 2 (SWING2) (Risi et al., 2012). $\delta^2 H_{seawater}$ in MPI-ESM-wiso has been calculated from $\delta^{18}O_{seawater}$ and d-excess outputs*".

l 210: "better reproduce isotopic change above 80°N than in the other models": I cannot see this in Fig 2. There isn't any observation above 80°N in this Fig.

Following a minor comment below on the Figure 2 to make to model results co-locate with the observations, we now present in the revised manuscript a new figure for the zonal distribution. We adapted the text accordingly.

Fig 1: I don't think the map for $\delta^{17}O$ is useful, since it shows exactly the same as for $\delta^2 H$. The added value of $\delta^{17}O$ relative to other isotopes is already well summarized by $^{17}O$-excess in Fig 1d. In contrast, I think that it would be worth to show the model-observation comparison for temperature and precipitation, because these variables can help interpret some of the model biases for isotopes.

We agree and removed the $\delta^{17}O$ results (spatial distribution and model-data comparison) from the main text. Instead, we added an Appendix A for the $\delta^{17}O$ to show in a first figure the spatial

distribution of the isotopic composition in the atmosphere and ocean and the model results against the observations in a second figure.

Fig 2: were the model outputs co-located with the measurements? For a more rigorous comparison, it might be useful to do so.

In the first version of the manuscript, Fig. 2 were presenting the zonal distribution of the mean latitudinal isotopic composition. In the revised manuscript we present a zonal multi-model comparison of the $\delta^2 H_{precipitation}$ and d-excess by taking the model outputs that co-locate with the measurements as suggested. However, to get a clear representation, we do not show each datapoint for the set of models but present the polynomial regression curve instead. Note that for the d-excess and $^{17}O$-excess, we removed the outlier values located in coastal regions of Antarctica (as highlighted in the manuscript) from the figure to get a better zonal multi-model comparison of the isotopic composition. We also decided to not show the zonal $^{17}O$-excess distribution in the main manuscript because only two models are currently available and because the Taylor diagram and the $^{17}O$-excess model-data comparison give sufficient statistical information to discuss on the capacity of the models to reproduce the $^{17}O$-excess. The zonal $^{17}O$-excess distribution is however presented in this document (Figure 6).

We developed the text in the Section 3.1.1 about $\delta^2 H_{precipitation}$ and 3.2.2 about d-excess in the revised manuscript to refer to this new figure.

[Figure]

*Figure 6: Multi-model zonal (a) $\delta^2 H_{precipitation}$, (b) d-excess and (c) $^{17}O$-excess comparison. The model results (in color) are compared to observations (in grey). The different lines are polynomial regression curves for the model results that co-locate with the observations.*

Fig 2: it would be useful to have the same for d-excess and $^{17}O$-excess. More generally, it looks like there is a new figure style for each isotopic variable. It would help the reader to have more coherent figures between the different variables. e.g. zonal mean for $\delta^2 H$, d-excess and $^{17}O$-excess, same style of model-obs scatter plot for $\delta^2 H$, d-excess and $^{17}O$-excess, etc...

See response on the comment above.

If there are too many figures, I think Fig 3 is not so useful. The MWL is not a stringent test on the simulations.

Following your comment and the one of the reviewer 2, we removed the Figure 3 from the manuscript.

l 285-290, 301-310: maybe these paragraphs could be summarized by just noticing that the spatial pattern of $\delta^{17}O$ looks almost exactly the same as $\delta^{18}O$? The $^{17}O$-excess parameter is what bears the added value.

Following your comment and the one of the reviewer 2 to remove the figures for $\delta^{17}O$, we simplified the section 3.1.3 to focus on the $^{17}O$-excess results only. We moved the $\delta^{17}O$ results to the Appendix A "$\delta^{17}O$ isotopic composition".

l 311: "proxy" -> "variable". For present day, $\delta^{17}O$ is directly measured.

We changed the word proxy to variable.

Fig 9: same for d-excess?

We added this new figure in the manuscript in addition to the already existing $\delta^{2}H_{precipitation}$ zonal figure. Similarly to the response above, we selected the model outputs that co-locate with the measurements.

l 424: "relatively similar close to zero values" -> values close to 0h. Same problem l200

We made the change in the text.

l 443: remove "a better agreement... at least", because only the second part of the sentence is correct.

Done.

Fig A1: I think this figure should replace Fig 4 in the text, and the appendix text can be merged in the main text. Everything that could be seen in Fig 4 can be seen in A1.

We replaced the Figure 4 by the Figure A1 and merged the text in the Appendix with the main text.

Please check the reference list. Some articles cited in the text are missing, e.g. Werner et al 2011.

The reference of Werner et al. 2011 was already in the reference list but we double-checked to make sure that every cited article was listed.

**References**

[revised manuscript text omitted]

---

## Author Comment (AC2)

**Response to Reviewer 2**

We thank Reviewer 2 for the comments on the different aspects of the manuscript. We answer them below (in blue) and will make changes accordingly in the revised manuscript.

- **Major comments**

As I said in the introduction, one major problem of the complex ESMs is the computing time. From this perspective, iLOVECLIM is very useful for paleoclimate simulations. The drawback of this model is the rough spatial and time (?) resolutions. I think this aspect of iLOVECLIM should be more emphasized in the introduction. Still for the introduction, this is in my knowledge the first time that $^{17}$O-excess is modeled in a coupled atmosphere-ocean model. Until now, only the atmospheric model LMDZ-iso was able to simulate the $H_2^{17}O$ isotopologue (Risi et al., 2013). This should be clearly stated in the abstract and the introduction.

Thank you for this comment. We added a new paragraph in the introduction to synthetize previous work the work on isotope modelling with the climate models: "*Since the initial works of Joussaume et al. (1984) and Jouzel et al. (1987), much progress has been done in atmospheric general circulation models (AGCMs) (e.g. Hoffmann et al., 1998; Noone and Simmonds, 2002; Mathieu et al., 2002, Risi et al., 2010; Werner et al., 2011) that can simulate accurately the $\delta^{18}O$ of precipitation. The subsequent development of water isotopes modules in oceanic general circulation models (OGCMs) (Schmidt, 1998; Delaygue et al., 2000; Xu et al., 2012) opens the possibility for coupled simulations of present and past climates, conserving water isotopes through the hydrosphere (Schmidt et al., 2007; Zhou et al., 2008; Tindall et al., 2009; Werner et al., 2016; Cauquoin et al., 2019). In general, General Circulation Models (GCMs) have been used exclusively to simulate separately water isotopes in the atmospheric and oceanic components. Given the computing resources needed to run coupled climate models, applying intermediate complexity coupled climate models with water isotopes like iLOVECLIM to long-term palaeoclimate perspectives still appears quite suitable (e.g. Caley et al., 2014). It could allow to improve our understanding of the relationship between water isotopologues, second-order parameter (like d-excess) and climate over a broad range of simulated climate changes*".

With respect to the $^{17}$O-excess we also added the following text in the introduction to highlight that very few model simulate this proxy: "*Modelling the $^{17}$O-excess is still very challenging since it depends on complex processes that have to be properly reproduced in the climate models. To date, only the LMDZ4 model has included the $^{17}$O-excess (Risi et al., 2013). However, even if the processes that control the $^{17}$O-excess are more complex than those controlling the d-excess, the combination of the d-excess, $^{17}$O-excess and $^{18}$O could bring new information on the understanding of past changes in local temperature, moisture origin and conditions at the moisture source*".

We also clearly stated in the abstract that we present modelled $^{17}$O-excess results. This now reads: "*Following previous developments of $\delta^{18}O$ in the coupled climate model of intermediate complexity iLOVECLIM, we present here the implementation of the $\delta^2H$ and $\delta^{17}O$ water isotopes in the different components of this model, and calculate the d-excess. We also present results of modelled $^{17}$O-excess in the atmosphere and ocean, that was currently only available in the LMDZ4 model …… The modelled $^{17}$O-excess presents a too important dispersion of the*

*values in comparison to the observations and is not correctly reproduced in the model mainly because of the complex processes involved in the $^{17}$O-excess isotopic value*".

Still for the introduction, for which kind of paleoclimate applications $^{17}$O-excess is useful? More generally, a paragraph of the introduction should be a review of the paleoclimate studies (recent if possible) using of d-excess or $^{17}$O-excess. For d-excess, such recent studies exist like Landais et al. (2021). For $^{17}$O-excess, I do not see to be honest as the measurements can be challenging. However, the author should try to explain how the $^{17}$O-excess can be used, not only by just saying that it is proxy of the relative humidity over the ocean. This kind of context information is necessary because simulating d-excess and $^{17}$O-excess is very challenging.

Thank you for pointing out this aspect. In the revised version of the manuscript, we developed the $^{17}$O-excess paragraph in the introduction to provide some context on this proxy. The following text has been added in the introduction: "*The $^{17}$O-excess is commonly used in ice core based paleoclimate studies to give information on the relative humidity over the ocean (e.g. Landais et al., 2008, 2018; Risi et al., 2010; Steig et al., 2021). $^{17}$O-excess is controlled by kinetic fractionation during evaporation, and similarly to d-excess, very sensitive to empirical parameter determining the supersaturation in polar clouds (Winkler et al., 2012; Landais et al., 2012). Since influences of temperature or condensation altitude on $^{17}$O-excess are expected to be insignificant in contrast to d-excess, measurements of $^{17}$O-excess have an added value with respect to d-excess and can be used to disentangle the parameters (temperature, relative humidity) that affect the water isotopic composition. For example, Risi et al. (2010) shown that the different behaviors of d-excess and $^{17}$O-excess in polar regions could be related to fractionation processes along the distillation pathway form the evaporative source to polar region that affect more the d-excess than the $^{17}$O-excess, that record more the signal from low latitudes during surface evaporation. Modelling the $^{17}$O-excess is still very challenging since it depends on complex processes that have to be properly reproduced in the climate models. To date, only the LMDZ4 model has included the $^{17}$O-excess (Risi et al., 2013). However, even if the processes that control the $^{17}$O-excess are more complex than those controlling the d-excess, the combination of the d-excess, $^{17}$O-excess and $^{18}$O could bring new information on the understanding of past changes in local temperature, moisture origin and conditions at the moisture source*".

I expect to use this kind of models for diverse paleoclimate applications. But which ones are really possible with a reasonable confidence? Before really reading the paper, I thought it would have been great to not only simulate pre-industrial conditions but also another climate period further in the past. As it is not the case, I recommend to the authors to do a deeper evaluation of their simulation against present-day observations with more skill metrics like r2and root mean square errors, and a comparison of these metrics with the ones from other general circulation models (GCMs) when available. Moreover, the authors should show more clearly if the well-known isotope continental effect and the amount effect are well represented in iLOVECLIM, in comparison to observations and other isotope enabled GCMs (like they did for the latitudinal effect). Last but not least, the disagreement between model results and observations is explained by uncertainties in the latter several times in the manuscript (e.g., l. 264-265, 289-291, 272-273, 310, 320-321, 443-445). I think these are not very honest statements. Instead, I would formulate a more quantitative model-data comparison, which would help the readers to know for which paleoclimate applications and isotope effects

iLOVECLIM can be used. In this regard, the figures 4, 5, 6 and maybe 7 need to be changed or adapted.

Thank you for this comment. As also suggested by the reviewer 1 we added a new figure to summarize some skill metrics for our model results, for existing water isotopes-enabled models and for the observations. We also detailed the different continental, amount and temperature effects in our model by investigating then individually. We then compared them to existing models like LMDZ4 and ECHAM5-wiso. Please see response to the first reviewer in Section 1.1 for this aspect.

In this paper we only investigated the capacity of our model to reproduce the hydrogen isotopic composition, d-excess and $^{17}$O-excess for present day, as it is a development paper. The simulation of the isotopic composition under another past climate period is not within the scope of this paper. But this opens new possibilities to perform long-term transient simulations with a model equipped with the isotopes since iLOVECLIM has the possibility to run simulations over several thousands of years within several weeks/months. For example, paper like Caley et al. (2014) already investigated past changes in the modelled oxygen isotopic composition during a glacial-interglacial cycle. We added a sentence in the manuscript to emphasize this aspect: "*Given the computing resources needed to run coupled climate models, applying intermediate complexity coupled climate models with water isotopes such iLOVECLIM to future long-term palaeoclimate perspectives appear very promising. Paleoclimate simulations during the Holocene, Last Glacial Maximum or transient glacial/interglacial periods are the next logical step to compare model results against past isotopic composition records*".

As already reported by the first reviewer, the fractionation for evapotranspiration is not supposed be at the equilibrium. Or there is no fractionation, like in MPI-ESM-wiso, or a fractionation using a bulk formula is used for the bare soil evaporation (i.e., kinetic, see the equation 6 from Haese et al., 2013). The simplest way is to perform another simulation without such fractionation in order to see the impact of your equation 10 and hopefully to improve the modeled results. Just an extension of a couple of hundred simulations should be enough, I guess.

Thanks for the remark and discussion of these processes. The text that is referred to was built from Roche (2013). A careful examination of the model code as used in the simulation presented in this study assumes no fractionation during all land-related evaporation processes, contrary to what was stated in the previous version of the manuscript. We removed Equation 10 and corrected the text accordingly: "*If re-evaporation occurs on land, it is assumed to be at equilibrium (without fractionation)*".

Before reading in detail the paper, I have been astonished by the very high and low values of $^{17}$O-excess, as well as their variations from one grid cell to another, in Figure 1. This is especially the case in Antarctica. As these are averages of several years, I guess these jumps are even worse from one year to another or at monthly scale. Honestly, I am worried by these huge variations. It is completely fine to not be able to represent very well the $^{17}$O-excess in such models because it is an extremely hard task. If the authors cannot fix this issue, I would expect honest suppositions on the causes of the failure of iLOVECLIM in simulating $^{17}$O-excess, instead of pseudo-explanations related to the uncertainties of the observations only. In addition, I suggest deleting all references and plots related to δ$^{17}$O. δ$^{17}$O is not really used

in the literature and does not bring any new information compared to $\delta^2H$ (the spatial characteristics are similar for example). The important proxy here is $^{17}O$-excess.

We agree and removed the $\delta^{17}O$ results (spatial distribution and model-data comparison) from the main text. Instead, we added an Appendix A for the $\delta^{17}O$ to show in a first figure the spatial distribution of the isotopic composition in the atmosphere and ocean and the model results against the observations in a second figure.

Based on the Taylor diagram and the model-data comparison, we observe for the $^{17}O$-excess a low correlation coefficient for iLOVECLIM and a low negative correlation coefficient for LMDZ4 with respect to observations. The standard deviation and root mean square error is better for LMDZ4 than for iLOVECLIM (Fig. 1c), suggesting that our model does not correctly reproduce the $^{17}O$-excess and has a too important dispersion of the values, even if the trend is correct. We now clearly state that iLOVECLIM does not correctly reproduce the $^{17}O$-excess values and suggest that this is mainly because of the complex processes involved in this isotopic composition and because of difficulties in modelling the isotopic composition for area with very low humidity content (especially for Antarctic values). In comparison, LMDZ4 shows indeed much smoother variations and less dispersion than iLOVECLIM but presents a general trend that is the opposite to the one observed in the measurements suggesting that both models do not perfectly reproduce the $^{17}O$-excess, but probably for different reason.

[Figure]

Figure 1: Taylor diagram representing (a) $\delta^2H_{precipitation}$, (b) d-excess and (c) $^{17}O$-excess values for different climate models (iLOVECLIM, LMDZ4, ECHAM5-wiso, CAM, GISS and MIROC) without Antarctic values. The simulated values are plotted against the observations. The dotted curved line indicates the reference line (standard deviation of the observation) and the bold grey contours represent RMSE values.

Based on the new figures to evaluate the model metrics like model-observation correlation or RMSE, we now properly state in the revised manuscript that the $^{17}O$-excess is not correctly reproduce in iLOVECLIM.

- **Line by line comments**

Title: I would change the title a little bit because the novelty here is to model $^1H^2H^{16}O$ and $H_2^{17}O$, not the $^{18}O$. Moreover, iLOVECLIM models the isotopologues (i.e., molecules), not the atoms of hydrogen and oxygen.

We changed the title to "*Modelling water isotopologues ($^1H^2H^{16}O$, $^1H_2^{17}O$) in the coupled numerical climate model iLOVECLIM (version 1.1.5)*" to take into account this suggestion.

l. 14-15: is the simulation really under preindustrial conditions as the orbital year considered is 1950 and not 1850?

The insolation is taken from the year 1950 but the other boundary conditions are taken from the preindustrial.

l. 24: "Stable water isotopologues ($H_2^{16}O$, $H_2^{18}O$, $^1H^2H^{16}O$, $H_2^{17}O$), expressed hereafter in the usual d notation with respect to V-SMOW scale (Dansgaard, 1964), are important…"

Done.

l. 29: The term "however" sounds strange here.

We removed this term.

l. 53: not so new method.

We replaced "A new method" by "Another method".

l. 61: same as above, the studies are not so recent. So, remove the term "More recently".

Done.

l. 65: A paragraph could be written about the use of d-excess and $^{17}O$-excess for paleoclimate studies. See major comment.

Done.

l. 99-100: the authors say they present the equations for deuterium only, but then the equations of 17O are shown latter in the manuscript (equations 7 and 9). I would say instead that you introduce the equations for the heavy/light isotope ratios.

We modified the sentence accordingly.

Equation 4 is from Craig and Gorgon (1965).

We changed the reference in the text.

Section 2.3: please add the time steps of the atmosphere and ocean modules. Also, do all the results come from the 100-years simulation starting from the 5000-year spin-up simulation?

The atmospheric module has a timestep of 6 hours and the oceanic module has a daily timestep. We added these timesteps in the respective Sections 2.1 and 2.2 for the atmosphere and the ocean.
The 100 years simulation starting from the 5,000 years spin-up has been parametrized to display monthly outputs. So, it is only used to investigate the seasonal variations of the precipitation and isotopic composition in Section 3.1.4. The 5,000 years simulation has annual outputs only and is used for the rest of the manuscript. We specified in Section 2.3 that the 100 years simulation is only used to investigate the seasonal variations.

Section 2.4: I would also mention the results from other isotope enabled GCMs here or in a new subsection just after. In the former case, please rename the section appropriately.

We renamed the Section 2.4 into Observational data and water isotopes enabled GCMs. This section presents all the datasets used in the manuscript to compare with the model results. We also added a new paragraph to mention the other isotopes-enabled GCMs used in this manuscript that reads: "*To evaluate our model results against water isotopes-enabled GCMs, we used several model outputs: ECHAM5-wiso (Steiger et al., 2018), GISS (Schmidt et al., 2007), LMDZ4 (Risi et al., 2010, Risi et al., 2013), MIROC (Kurita et al., 2011), CAM (Lee et al., 2007) and MPI-ESM-wiso (Cauquoin et al., 2020). The GISS, LMDZ4, MIROC and CAM data are from the Stable Water Isotope Intercomparison Group, Phase 2 (SWING2) (Risi et al., 2012). $\delta^2 H_{seawater}$ in MPI-ESM-wiso has been calculated from $\delta^{18}O_{seawater}$ and d-excess outputs*".

l. 177 and many others: I do not understand the reference IAEA, 2006. All GNIP data should be mentioned with the reference IAEA, 2023.

We corrected the reference to IAEA, 2023.

l. 181-182: why the authors did choose these stations, and not others like Vienna? What are the requirements (e.g., number of consecutive years with data)? How did they make the composite (I mean on which period or on how many years)?

We chose specific stations that are representative of various climate conditions (northern Atlantic, eastern Mediterranean, South Africa and South America). We could have used Vienna station like presented in Figure 2 below, as any other station where the isotopic composition has been reported for a minimum of 3 calendar years within the period 1961-2008.
To investigate these monthly variations, we used the 100 years model simulation. We then kept the last 10 years and calculated the seasonal mean over this time period. For easier comparison with the data, we normalized the data by subtracting the annual mean and dividing by the standard deviation for each station. This has been added to the main text.

[Figure]

*Figure 2: Monthly evolution of the precipitation (left), $\delta^2 H_{precipitation}$ (middle) and d-excess (right) for Vienna station. The red line is the GNIP data and the blue line is the iLOVECLIM model. The data have been normalized. The error bars for the data are also shown at 2σ.*

l. 190-191: You already said in the data section which dataset you will use for the evaluation of your results. You do not need to repeat here again.

We removed it from the text.

l. 193: Please rephrase "Differences with the observations are observed for specific regions.".

We changed the sentence to "*Regions like central Africa and northern region of South America show however differences with the data since the modelled $\delta^2 H_{precipitation}$ is underestimated in comparison to the few measurements available*".

l. 204-205: I suppose these model results are from SWING2 database. Please add the reference (Risi et al., 2012) and state it clearly.

Yes you are right. We specified that the model outputs used in this study (except the MPI-ESM-wiso and ECHAM5-wiso) come from the SWING2 database and added the reference of Risi et al. (2012).

l. 206: such as strong depletions over Antarctica?

We corrected the sentence.

l. 208- 209: "Similarly to other GCMs, iLOVECLIM shows a small decrease of d2Hprecipitation and is in the higher range of the observed δ2Hprecipitation values."

We modified the sentence accordingly.

Sentence at l. 209-210: I do not understand this sentence and it should be removed.

We removed this sentence.

Figure 1 and all the other concerned figures: remove the $\delta^{17}O$, it's not useful, I think.

We moved the $\delta^{17}O$ figures from the main text to the Appendix A.
l. 231-232: please precise what could be these complex processes.

We added the following at the end of the sentence: "*such as the behaviour of the advection scheme at very low moisture content or the role of kinetic fractionation coefficient*".

Figure 3: is it really useful? I think this figure can be removed.

We removed the figure.

l. 246-247: same comment as for l. 190-191.

Done

l. 253: you say that the model calculates mostly negative values with values ranging from -10 to 10 permil. It sounds a little bit strange, no?

*Yes we agreed. We changed the sentence to "the model calculates values ranging from -10 to 25 ‰".*

l. 264-265: see my main comment about a fair evaluation of your model results.
l. 272-273: same comment.
l. 289-291: same comment as for l. 190-191. Please explain the possible causes in terms of model biases.
l. 311-313: see major comment about a fair evaluation of the model.
l. 320-321: same comment.

*Based on the new figures to evaluate the amount effect, temperature effect, continental effect and to calculate metrics for iLOVECLIM and other GCMs in comparison to measurements (see response to the first reviewer in Section 1.1), we corrected the text in the manuscript to fairly evaluate the model results against other isotopes-enabled models and the observations.*

l. 307: $H_2^{17}O$ instead of $^{17}O$.

*Done.*

Section 3.1.4: Why these stations in particular? I know that $^{17}O$-excess is not available in GNIP data (and it should be stated). Is there any data of $^{17}O$-excess in precipitation or in water vapor at seasonal resolution (at least) to evaluate iLOVECLIM? Moreover, the evaluation should be done in a fairer way (again). The uncertainties of the data alone do not explain the model-data disagreements.

*We chose stations that are representative of various climate conditions (see previous response). We added a sentence in the main text to precise that $^{17}O$-excess data are not available in the GNIP database.*

Section 3.2 should be re-organized a little bit for clarification. You can also make separate sub-sections for d-excess and $^{17}O$-excess. Moreover, even if there are no observations 17O-excess in deep ocean, I would expect to see the results from iLOVECLIM because this is one novelty of this model.

*Thank you for this suggestion. We reorganized the Section 3.2 in a new Section 3.3 Isotopes in ocean water, that is separated in two sub-sections for surface seawater and vertical profiles. We also added a new panel on the existing figure of the isotopic depth distribution, to represent the modelled $^{17}O$-excess (Figure 3). With respect to this figure, we added the following text in the revised manuscript: "The oceanic d-excess and $^{17}O$-excess shows less prominent influence of the main water masses. Above 1000 m, the d-excess goes from 40°S to 40°N with depleted negative values, and enriched positive values for $^{17}O$-excess. Below 1000 m and from 40°S to the north, the NADW d-excess values are higher with a maximum of 2 ‰ around 25°N and 2000 m depth. On the opposite, $^{17}O$-excess values are lower than in the surface, with minimum values at the same latitude and depth than d-excess. The comparison with the $\delta^2H$ and d-excess observations shows that the model reproduces the depleted surface values and the enriched d-excess values below 1800 m even if the latitudinal gradient is more pronounced in the model than in the data. The depth interval from 500 to 1800 m presents a disagreement between the modelled d-excess and the observation values that are consistently lower than in*

*the model. This is especially the case for high latitudes of the northern hemisphere where the difference between the model and the data can reach 2 to 3 ‰. Since no $^{17}$O-excess observations exist at depth, we refrain for any further evaluation of the modelled values*".

[Figure]

*Figure 3: Atlantic zonal mean in iLOVECLIM of (a) $\delta^2$H of seawater, (b) d-excess of seawater and (c) $^{17}$O-excess of seawater compared to observations.*

l. 361-362: you should say that in the observation data section.
l. 370-371: it should be in data section.

We moved these two parts to the Section 2.4 Observational data and water isotopes enabled GCMs.

l. 371 and 372: replace MPI-ESM by MPI-ESM-wiso. Do it also in the legend of the concerned figures.

Done.

l. 387-388: It's one explanation. Usually, very depleted $\delta^{18}$O or $\delta^2$H values in seawater in Artic area are explained by the very depleted river discharges. What about iLOVECLIM? If it is not modelled, it is one very plausible explanation for this bias.

iLOVECLIM does not model the river discharges. So it could indeed be one explanation for the enriched isotopic values obtained in the model. We added this hypothesis in the main text.

l. 406: I would say instead that model d2H values are lower than the observations by several permil.

We changed the sentence accordingly.

l. 434: "we presented the implementation of the $^1H^2H^{16}O$ and $H_2^{17}O$ isotopologues in the …"

Done.

End of line 435: remove "also".

Done.

l. 439-440 and 443-445: see main comment about the evaluation of iLOVECLIM results.

Following your previous comments, we changed the text of the conclusion that now reads: "*For the atmospheric part, we found a good agreement between the model, the GNIP data (considering the intrinsic biases of iLOVECLIM that could lead to local inconsistencies) and several GCMs, with the conservation of the latitudinal gradient. The modelled $\delta^2H$ and $\delta^{18}O$ also fit with the global Meteorological Water Line and the main isotopic effect (amount effect, temperature effect and continental effect are well reproduced in the model). The d-excess distribution for the atmosphere is also correctly modelled at global scale in comparison to the observations and several GCMs. The isotopic composition of oxygen and hydrogen over Antarctica present however differences of several permil in comparison to the data because of the complexity of the local processes at play that are simplified in the model. At present, our models-data comparison suggests that iLOVECLIM does not correctly reproduce the $^{17}O$-excess and has a too important dispersion of the values. Modelling the $^{17}O$-excess has to be improved in the future versions of the isotopes-enabled models. New measurements are also needed with a reduction of their associated uncertainties*".

Figure A1: it should be in the main text.

We moved this figure and related text in the Section 3.1.2.

**References**

[revised manuscript text omitted]

---

## Referee Report (RR1)

**Review of Extier et al**

December 14, 2023

The paper has been improved by the revisions. I still have many comments. I think they can be fixed easily, but they are numerous.

- l 12: "$\delta^2 H$ and $\delta^{17}O$ water isotopes" -> "$^1H^2H^{16}O$ and $^1H_2^{17}O$ water isotopes"

- l 18: "Models and the..." -> "Models. The main isotopic effects and the latitudinal gradient are properly modeled, similarly to previous water isotope-enabled General Circulation Models."

- l 32: "the $\delta^2 H$ and $\delta^{18}O$ isotopic ratios of precipitation" -> "the $\delta^2 H$ and of precipitation"

- l 32: "ice cores" -> "polar ice cores" (things are different in tropical ice cores)

- l 34: ", and following ... latitudes": remove, I don't understand what it means.

- l 50: "experiment" -> "experimental"

- l 74: "plant lipids wax" -> "plant lipids" or "plant wax"

- l 76: "waters" -> "water"

- l 82: missing empty line

- l 85: "allows estimations of past regional and qualitative changes" -> "allows qualitative estimations of past regional changes"

- l 93: remove Risi et al 2010, which was not coupled. Rather look for HadCM or GISS references.

- l 99: "works" -> "work"

- l 107: "still appears quite suitable" -> "is suitable"

- l 107: add a sentence on other existing isotope-enabled intermediate complexity models: CLIMBER? Speedier?

- l 125: clarify what are the layers: from which level to which level? Are 800, 500 and 200 hPa the middle of the layers?

- l 138: Merlivat and Jouzel 1979 reference here is out of place. They don't say they look at precipitable water.

- l 138: recall that the moisture is assume to be only in the first layer.

- l 147: this part is still not clear. All isotope-enabled GCMs adopt the same kind of equation to calculate $R_E$, even though there is vertical discretization of water and isotopes in these models. So I don't think the lack of vertical discretization in iLOVECLIM is what justifies the formula for $R_E$. I advice to clarify what are the consequences of the lack of vertical discretization. e.g. is there a systematic bias in $R_a$? "solution adopted by Roche (2013)" is misleading, since the same formula was used in all isotope-enabled GCMs, already long before 2013.

- l 164: clarify the 3 types of precipitation: e.g. how about large-scale snow? Do you mean convective rain, large-scale rain, and snow?

- l 165: "values at 650, 800 and 650 hPa": it was previously written that there was no vertical discretization for water isotopes? Please clarify.

- l 166: "fractionation schemes" -> "fractionation coefficients"?

- l 166: "fractionation" -> "equilibrium fractionation"? same l 168?

- l 167: "enhanced kinetic fractionation at high latitude": do you mean the supersaturation effect? If so, please clarify this, and replace Merlivat and Jouzel 1979 by [Ciais and Jouzel, 1994].
  General: How do you account for the kinetic effect associated with the supersaturation at cold temperature? Do you use a linear function of supersaturation as a function of temperature like in all GCMs? Please explain.

- l 166-168: this sentence is really not clear. Please replace it by a clear equation, or remove.

- l 227: "annual mean" -> "annual-mean"
  Same l 506.

- Fig 2: make text in the keys larger. Same Fig 3

- Fig 3: precise if the values are monthly values. If the case, it represents both spatial and seasonal variations

- l 277: "with a correlation coefficient of ... 0.99" -> "with a correlation coefficient of 0.99"

- l 303: "and could be used...": I would replace by ". The same caution should be required for iLOVECLIM as for other GCMs when investigating past changes in d-excess."

- Fig 4 is wrongly named Fig 2; problem with the numbering of all figures starting here.

- Discussion around Fig 4: in GCMs, d-excess in high latitudes is very sensitive to the parameterization of supersaturation. Is it also the case here? Or are temperatures not cold enough?

- l 356: "even if ... fit" -> "with most of the data fitting"

- l 402: "composition" -> "ratio". General rule: the ratio is a number, the composition is a qualitative property.

- l 404: " to see if ... precipitation." -> "because this is where the amount effect is observed."

- l 411: "secondary evaporation": what does this mean? Does it mean the rain evaporation? But it was written this process is ignored in iLOVECLIM?

- l 414: "delay" -> "advance"?

- The fact that iLOVECLIM can simulate the amount effect deserves to be discussed. [Lee and Fung, 2008, Risi et al., 2008, Risi et al., 2021] show the key role of rain evaporation in the amount effect. [Field et al., 2010] even shows that disabling the fractionation during rain-vapor interactions suppressed the amount effect in a GCM. The capacity of iLOVECLIM to simulate the amount effect without this fractionation is thus surprising. In contrast, several studies give an integrated water budget perspective to the amount effect [Lee et al., 2007, Moore et al., 2014], which could explain the capacity of iLOVECLIM to simulate the amount effect.

- Fig 8: why normalizing the values? Is this hiding a problem with the amplitude in iLOVECLIM? I would advice to show the real value, for transparency.

- Fig 7: Why is LMDZ alone on its plot? Why is the x-axis unit different for the two plots? I suggest to use the same precipitation unit for all observations and models and plot everything on the same plot. If too busy, then add observations on each plot as a reference.

- l 441: again, what does 650 hPa mean? Is this an interlayer level?

- sec 3.2.2: why comparing iLOVECLIM with only one GCM? It's OK but needs to be justified: e.g. is LMDZ representative of all other GCMs?

- LMDZ is too enriched at cold temperatures with respect to observations, for reasons given in [Cauquoin et al., 2019]

- l 458: "with fractionation during continental recycling": no! even without fractionation during continental recycling, the continental effect is observed, as shown by all isotope-enabled GCMs. It is due to the fact that over land, the enrichment of the low-level vapor by evaporation is weaker than over the ocean. Over land, not all the precipitation goes back to the atmosphere, so heavy isotopes are preferentially lost by runoff (e.g. [Pierrehumbert, 1999] for a simple model of this effect). The fractionation during bare soil evaporation only very slightly enhanced the continental effect [Haese et al., 2013, Risi et al., 2016]. For the observed continental effect, cite [Rozanski et al., 1993].
  l 470: ""Even if... fractionation": remove, since the fractionation during bare soil evaporation is not responsible for the continental effect.

- Why documenting the continental effect in the tropics? It is largest at mid and high latitudes. In the tropics, it is weak [Salati et al., 1979, Worden et al., 2021] or even reversed (more enriched over land) [Levin et al., 2009], due to strong evapo-transpiration. I would have expected the same plot for mid and high latitudes.

- l 478: "more complex": is it really more complex than in LMDZ? Any reference to justify this assertion? The main difference between LMDZ and ECHAM seems to be the horizontal resolution, not the complexity of its parameterization.

- l 515: I don't understand the logic: why the "absence of sea ice ... would lead to fractionation during sea ice formation"?

- l 518: "more depleted d-excess" -> "lower d-excess"
  l 521: "as well highly depleted" -> "very low"
  General rule: the vapor is depleted, but the $\delta^{18}O$ , the d-excess or the $^{17}O$-excess are low.

- l 544: "observation" -> "observations" or "dataset"

- l 560: "than d-excess" -> "as the d-excess maximum"

- l 573: "with the conservation of" -> "with a reasonable simulation of"

- l 579: "that has a too important" -> "with an excessive"

- l 580: "isotopes-enabled" -> "isotope-enabled"

- l 581: "a good accordance of" -> "with good agreement"

**References**

[Cauquoin et al., 2019] Cauquoin, A., Risi, C., and Vignon, É. (2019). Importance of the advection scheme for the simulation of water isotopes over antarctica by atmospheric general circulation models: A case study for present-day and last glacial maximum with lmdz-iso. *Earth and Planetary Science Letters*, 524:115731.

[Ciais and Jouzel, 1994] Ciais, P. and Jouzel, J. (1994). Deuterium and oxygen 18 in precipitation: isotopic model, including cloud processes. *J. Geophys. Res.*, 99:16,793–16,803.

[Field et al., 2010] Field, R. D., Jones, D. B. A., and Brown, D. P. (2010). The effects of post-condensation exchange on the isotopic composition of water in the atmosphere. *J. Geophy. Res.*, 115, D24305:doi:10.1029/2010JD014334.

[Haese et al., 2013] Haese, B., Werner, M., and Lohmann, G. (2013). Stable water isotopes in the coupled atmosphere-land surface model ECHAM5-JSBACH. *Geoscientific Model Development*, 6:1463–1480, doi: 10.5194/gmd–6–1463–2013.

[Lee and Fung, 2008] Lee, J.-E. and Fung, I. (2008). "Amount effect" of water isotopes and quantitative analysis of post-condensation processes. *Hydrological Processes*, 22 (1):1–8.

[Lee et al., 2007] Lee, J.-E., Fung, I., DePaolo, D., and Fennig, C. C. (2007). Analysis of the global distribution of water isotopes using the NCAR atmospheric general circulation model. *J. Geophys. Res.*, 112:D16306, doi:10.1029/2006JD007657.

[Levin et al., 2009] Levin, N. E., Zipser, E. J., , and Cerling, T. E. (2009). Isotopic composition of waters from Ethiopia and Kenya:Insights into moisture sources for eastern Africa. *J. Geophys. Res.*, 114:D23306, doi:10.1029/2009JD012166.

[Moore et al., 2014] Moore, M., Kuang, Z., and Blossey, P. N. (2014). A moisture budget perspective of the amount effect. *Geophys. Res. Lett.*, 41:1329–1335, doi:10.1002/2013GL058302.

[Pierrehumbert, 1999] Pierrehumbert, R. T. (1999). Huascaran delta18O as an indicator of tropical climate during the Last Glacial Maximum. *Geophys. Res. Lett.*, 26:1345–1348.

[Risi et al., 2016] Risi, C., Bony, S., Ogée, J., Bariac, T., Raz-Yaseed, N., Wingate, L., Welker, J., Knohl, A., Kurz-Besson, C., Leclerc, M., Zhang, G., N, B., Santrucek, J., Hronkova, M., David, T., Peylin, P., and Guglielmo, F. (2016). The water isotopic version of the land-surface model ORCHIDEE: implementation, evaluation, sensitivity to hydrological parameters. *Hydrology: Current Research*, 7:DOI: 10.4172/2157–7587.1000258.

[Risi et al., 2008] Risi, C., Bony, S., and Vimeux, F. (2008). Influence of convective processes on the isotopic composition (O18 and D) of precipitation and water vapor in the Tropics: Part 2: Physical interpretation of the amount effect. *J. Geophys. Res.*, 113:D19306, doi:10.1029/2008JD009943.

[Risi et al., 2021] Risi, C., Muller, C., and Blossey, P. (2021). Rain evaporation, snow melt, and entrainment at the heart of water vapor isotopic variations in the tropical troposphere, according to large-eddy simulations and a two-column model. *J. Adv. Model. Earth Sci.*, 13(4):e2020MS002381, DOI: https://doi.org/10.1029/2020MS002381.

[Rozanski et al., 1993] Rozanski, K., Araguas-Araguas, L., and Gonfiantini, R. (1993). Isotopic patterns in modern global precipitation. *Geophys. Monogr. Seri., AGU*, Climate Change in Continental Isotopic records.

[Salati et al., 1979] Salati, E., Dall'Olio, A., Matsui, E., and Gat, J. (1979). Recycling of water in the Amazon basin: An isotopic study. *Water Resources Research*, 15:1250–1258.

[Worden et al., 2021] Worden, S., Fu, R., Chakraborty, S., Liu, J., and Worden, J. (2021). Where does moisture come from over the congo basin? *Journal of Geophysical Research: Biogeosciences*, 126(8):e2020JG006024.

---

## Author Response (AR2)

**Response to Reviewer 1**

We thank Reviewer 1 for the comments and suggestions that helped to improve the new manuscript. We addressed all the comments of the referee (responses in blue) and changed the manuscript accordingly.

- l 12: $\delta 2H$ and $\delta 17O$ water isotopes -> $1H2H16O$ and $1H172O$ water isotopes

We changed it in the text.

- l 18: Models and the... -> Models. The main isotopic effect and the latitudinal gradient are properly modeled, similarly to previous water isotope-enabled General Circulation Models.

The new text now reads: "*The main isotopic effects and the latitudinal gradient are properly modelled similarly to previous water isotopes-enabled General Circulation Models simulations despite a simplified atmospheric component in iLOVECLIM*".

- l 32:  the $\delta 2H$ and $\delta 18O$ isotopic ratios of precipitation -> the $\delta 2H$ and of precipitation

Done.

- l 32: ice cores -> polar ice cores (things are different in tropical ice cores)

We corrected it in the text.

- l 34: , and following ... latitudes  : remove, I don't understand what it means

Done.

- l 50:  experiment -> experimental

Done.

- l 74:  plant lipids wax -> plant lipids or plant wax

We changed it to plant wax.

- l 76:  waters -> water

Corrected.

- l 82: missing empty line

We added a line to separate the paragraph.

- l 85: allows estimations of past regional and qualitative changes -> allows qualitative estimations of past regional changes

We changed the sentence accordingly.

- l 93: remove Risi et al 2010, which was not coupled. Rather look for HadCM or GISS references.

We replaced the reference of Risi et al. (2012) by Schmid et al. (2007) for GISS model, and by Tindall et al. (2009) for HadCM model.

- l 99:  works -> work

Done.

- l 107:  still appears quite suitable -> is suitable

Done.

- l 107: add a sentence on other existing isotope-enabled intermediate complexity models: CLIMBER? Speedier?

We adapted the text that now reads: "*Other isotopes-enabled intermediate complexity models exist like CLIMBER (Roche et al., 2004), or fast GCM like SPEEDY-IER (Dee et al., 2015), that could be used to improve our understanding of the relationship between water isotopologues, second-order parameter (like d-excess) and climate over a broad range of simulated climate changes*".

- l 125: clarify what are the layers: from which level to which level? Are 800, 500 and 200 hPa the middle of the layers?

We detailed the text that now reads: "*It is subdivided in three vertical layers: (1) between the surface and 650 hPa (2) between 650 and 350 hPa and (3) between 350 and 0 hPa. 800, 500 and 200 hPa are respectively the mid-point of each layer*".

- l 138: Merlivat and Jouzel 1979 reference here is out of place. They don't say they look at precipitable water.

We removed the reference from this paragraph.

- l 138: recall that the moisture is assume to be only in the first layer.

We added this clarification to the text.

- l 147: this part is still not clear. All isotope-enabled GCMs adopt the same kind of equation to calculate RE, even though there is vertical discretization of water and isotopes in these models. So I don't think the lack of vertical discretization in iLOVECLIM is what justifies the formula for RE. I advice to clarify what are the consequences of the lack of vertical discretization. e.g. is there a systematic bias in Ra? solution adopted by Roche (2013) is misleading, since the same formula was used in all isotope-enabled GCMs, already long before 2013.

We agree that the solution adopted is very similar to what is used in GCMs. The originality of the formulation of Roche (2013) resides in the use of the apparent humidity $h^*_a$ which is a surrogate consequent to the lack of vertical discretization. Regarding the values of $R_a$, it is not obvious at present how to recompute an equivalent field from GCM outputs or reanalyses given that in ECBilt this variable represents the isotopic ratio of the total water content of the atmosphere at 850 hPa. However, given that the large-scale results in isotopic content in precipitation (Figure 2 and 3 in the manuscript) is comparable to other GCMs, there should not be a massive bias.

- l 164: clarify the 3 types of precipitation: e.g. how about large-sale snow? Do you mean convective rain, large-sale rain, and snow?

We clarified this sentence to say that we are referring about convective rain large-scale rain and snow.

- l 165: values at 650, 800 and 650 hPa: it was previously written that there was no vertical discretization for water isotopes? Please clarify.

As specified in the ECBilt model description, the model is discretized in three vertical layers (including temperature which is what we refer to in the sentence mentioned), only water content is confined to the first layer.

- l 166: fractionation schemes -> fractionation coefficients?

We replaced the word in the main text.

- l 166: fractionation -> equilibrium fractionation? same l 168?

We corrected it in the main text.

- l 167: enhanced kinetic fractionation at high latitude: do you mean the supersaturation effect? If so, please clarify this, and replace Merlivat and Jouzel 1979 by [Ciais and Jouzel, 1994]. General: How do you account for the kinetic effect associated with the supersaturation at cold temperature? Do you use a linear function of supersaturation as a function of temperature like in all GCMs? Please explain.

Thanks for the comment. We removed the misleading sentence in the paragraph. Currently there is no kinetic fractionation associated with supersaturation at low temperatures.

- l 166-168: this sentence is really not clear. Please replace it by a clear equation, or remove.

See response just above.

- l 227: annual mean -> annual-mean

Same l 506.

We corrected every iteration of annual-mean in the text.

- Fig 2: make text in the keys larger. Same Fig 3

We increased the size of the text for Figures 2 and 3 in the manuscript.

- Fig 3: precise if the values are monthly values. If the case, it represents both spatial and seasonal variations

The values presented in Figure 3 are annual values.

- l 277: with a correlation coefficient of ... 0.99 -> with a correlation coefficient of 0.99

Done.

- l 303: and could be used...: I would replace by ". The same caution should be required for iLOVECLIM as for other GCMs when investigating past changes in d-excess."

Thank you for pointing out this sentence. We modified it in the revised manuscript.

- Fig 4 is wrongly named Fig 2; problem with the numbering of all figures starting here.

We corrected the numbering of the figures in the revised manuscript.

- Discussion around Fig 4: in GCMs, d-excess in high latitudes is very sensitive to the parameterization of supersaturation. Is it also the case here? Or are temperatures not cold enough?

Figure 4 clearly shows that our simulated range of $\delta^2H_{precipitation}$ is not low enough compared to the data. In the range simulated, the effect of supersaturation remains small.

- l 356: even if ... fit -> with most of the data fitting

We removed this text according to the new paragraph to detail the new Figure 6 in the revised manuscript.

- l 402: composition -> ratio. General rule: the ratio is a number, the composition is a qualitative property.

We changed it in the text.

- l 404: to see if ... precipitation. -> because this is where the amount effect is observed.

We modified the sentence.

- l 411: secondary evaporation: what does this mean? Does it mean the rain evaporation? But it was written this process is ignored in iLOVECLIM?

Thanks for pointing it out. We removed the sentence.

- l 414: delay -> advance?

We changed the sentence that now reads: "*A lag of one month is also observed between the data and LMDZ4 for the north tropics*".

- The fact that iLOVECLIM can simulate the amount effect deserves to be discussed. [Lee and Fung, 2008, Risi et al., 2008, Risi et al., 2021] show the key role of rain evaporation in the amount effect. [Field et al., 2010] even shows that disabling the fractionation during rain-vapor interactions suppressed the amount effect in a GCM. The capacity of iLOVECLIM to simulate the amount effect without this fractionation is thus surprising. In contrast, several studies give an integrated water budget perspective to the amount effect [Lee et al., 2007, Moore et al., 2014], which could explain the capacity of iLOVECLIM to simulate the amount effect.

Thank you for providing these inputs, we were unaware of the dispute in the GCMs community about the origin of the amount effect. The arguments presented to interpret the results in the complex model of Risi et al. (2021) are far beyond any process that are modelled in our simplified atmospheric model. This is also in part true for the GCM results of Moore et al. (2014). Our approach is much simpler. The amount effect depletion for us is the process related to sequential precipitation removal and under-replenishment, in the form identified early by Dansgaard (1964). How this relates to the very detailed processes described in Risi et al. (2021) is hard to figure out and certainly beyond the scale of our current study. We would agree with the reviewer that in a way, our simple approach is more comparable to that of Moore et

al. (2014), but this would need some confirmation using site specific budget in the model identifying the terms of the budget of Moore et al. (2014), provided that all are accessible in our model (some are clearly not like a separate isotopic ratio of the boundary layer). The resolution of the dispute would not, in any case, change what is currently presented in the manuscript since we computed the iLOVECLIM and GCMs result in the same manner, thus comparing like with like from simple isotopic mean budgets

- Fig 8: why normalizing the values? Is this hiding a problem with the amplitude in iLOVECLIM? I would advice to show the real value, for transparency.

We chose to normalize the values because the seasonal evolution of precipitation and isotopic ratio in the model is not expected to perfectly reflect the measurements, as mentioned in the previous version of the manuscript L377-378.

We present here the raw values in Figure 1. As seen in this figure, the seasonal variation of the precipitation and $\delta^2 Hp_{recipitation}$ is the same than the one presented in the manuscript with the normalized values (Figure 8). The lead and lag of the two models compared to the data is also conserved. Differences are however observed in the amplitude, mostly for the isotopic ratio, with lower values up to 15 ‰ in summer for the north tropics between the data and the models. Same difference in absolute values between the observation and the models is observed in the south tropics. We then keep in the revised manuscript figures with normalized values to account for the fact that the seasonal evolution of precipitation and isotopic ratio in the model is not expected to perfectly reflect the measurements.

[Figure]

Figure 1: Seasonal variations of the mean precipitation and $\delta^2 H_{precipitation}$ in the tropics, from 0-20°N for (a) and from 0-20°S for (b). The solid lines represent the precipitation and the dashed lines the $\delta^2 H_{precipitation}$. The blue curve presents the iLOVECLIM values, the red curve is for LMDZ4 and the green curve corresponds to the GNIP data.

- Fig 7: Why is LMDZ alone on its plot? Why is the x-axis unit different for the two plots? I suggest to use the same precipitation unit for all observations and models and plot everything on the same plot. If too busy, then add observations on each plot as a reference.

LMDZ4 was presented on a separated plot because the units for precipitation were in kg/s.m$^2$, whereas it was in cm/y for iLOVECLIM and GNIP data. Considering that 1 kg of rain water spread over 1 square meter of surface is 1 mm in thickness, we converted the LMDZ4 results to have the same unit than iLOVECLIM and the observations for easier comparison. Results are presented on Figure 2, that replaces the Figure 9 in the revised manuscript. We now

compare the amount effect between the two models and the observations (-0.085‰/cm.y⁻¹ in iLOVECLIM, -0.103‰/cm.y⁻¹ in LMDZ4 and -0.139‰/cm.y⁻¹ for the observations).

[Figure]

Figure 2: Monthly δ²H$_{precipitation}$ as a function of the precipitation at the location of nine tropical oceanic GNIP stations. iLOVECLIM results in blue are compared to LMDZ4 in red and to GNIP data in green. The error bars for the data are shown at 2σ.

- l 441: again, what does 650 hPa mean? Is this an interlayer level?

We specified that 650 hPa corresponds to the top of the first layer.

- sec 3.2.2: why comparing iLOVECLIM with only one GCM? It's OK but needs to be justified: e.g. is LMDZ representative of all other GCMs?

In this section we decided to compare iLOVECLIM results only with LMDZ4 because the Taylor diagram and the multi-model zonal comparison (Figures 2 and 3 in the manuscript) show that the LMDZ4 isotopic composition is within the range of other GCMs. So we consider that LMDZ4 is representative of all other GCMs.

- LMDZ is too enriched at cold temperatures with respect to observations, for reasons given in [Cauquoin et al., 2019]

Following your comment, we added a sentence to specify the reason for too enriched isotopic values in LMDZ: "*As shown in Cauquoin et al. (2019), the representation of the advection scheme in the model can impact the isotopic composition, with more enriched values when a more diffusive advection scheme is applied*".

- l 458: with fractionation during continental recycling: no! even without fractionation during continental recycling, the continental effect is observed, as shown by all isotope-enabled GCMs. It is due to the fact that over land, the enrichment of the low-level vapor by evaporation is weaker than over the ocean. Over land, not all the precipitation goes back to the atmosphere, so heavy isotopes are preferentially lost by runoff (e.g. [Pierrehumbert, 1999] for a simple model of this effect). The fractionation during bare soil evaporation only very slightly enhanced the continental effect [Haese et al., 2013, Risi et al., 2016]. For the observed continental effect, cite [Rozanski et al., 1993].

We modified the text of the Section 3.2.3 following the suggestion made.

- l 470: Even if... fractionation: remove, since the fractionation during bare soil evaporation is not responsible for the continental effect.

We removed this sentence in the main manuscript.

- Why documenting the continental effect in the tropics? It is largest at mid and high latitudes. In the tropics, it is weak [Salati et al., 1979, Worden et al., 2021] or even reversed (more enriched over land) [Levin et al., 2009], due to strong evapo-transpiration. I would have expected the same plot for mid and high latitudes.

Following comment from the Reviewer 1 in the initial review, we looked at the continental effect in the tropics. We however agree that this continental effect is observed and better seen at high latitudes. Based on the same methodology than presented in the first response to the reviewers, we present here a new figure for the continental effect for regions between 40-70°N. We chose to focus on the northern hemisphere since most of the land area are located there, and the continental effect is the most visible. We compared the GNIP data with iLOVECLIM, LMDZ4 and ECHAM5-wiso, with the number of stations and points in the model summarized in the Table 1 below. Note that very few stations are available in the observations for the ocean. We then calculated series of mean values corresponding to the continents (Europe, Asia and North America) and to the oceans (Atlantic, Pacific, Arctic), and added the results to the existing Figure 11 in the revised manuscript (Figure 3 below).

| | 40-70°N | |
| --- | --- | --- |
| | Continent | Ocean |
| GNIP | 107 | 4 |
| iLOVECLIM | 278 | 174 |
| LMDZ4 | 766 | 357 |
| ECHAM5-wiso | 7853 | 4178 |

Table 1: Number of GNIP stations and points in the different models that cover land surfaces and oceans between 40-70°N.

The contrast in isotopic value between land and ocean observed in the data and models, with more depleted values over land is due to the fact that over land, the enrichment of the low-level vapor by evaporation is weaker than over the ocean. This is well observed between 40-70°N in the observations with a median value of -89.8 ‰ for the continents and -51 ‰ for the oceans (Figure 3e). This continental effect is also observed in iLOVECLIM, LMDZ4 and ECHAM5-wiso with respective values of -52 ‰, -99.8 ‰ and -109.8 ‰ for the continents and -31.3 ‰, -43.2 ‰ and -59.5 ‰ for the oceans (Figure 3f, g, h). The amplitude of the continental effect for these mid to high latitudes is less pronounced in iLOVECLIM than in the observations (-20.7 ‰ vs -38.9 ‰), as observed for the tropics. In comparison, LMDZ4 and ECHAM5-wiso models have higher continental effect than observations (-56.6 ‰ and − 50.3 ‰ vs -38.9 ‰).

The continental effect is also less pronounced at low latitudes than at mid-high latitudes. This comparison has been added to the revised manuscript.

[Figure]

*Figure 3: Box plots of the δ²H$_{precipitation}$ over the continents (in green) and oceans (in blue). The panels (a) to (d) present values between 0-20°N and 0-20°S for (a) GNIP data, (b) iLOVECLIM, (c) LMDZ4 and (d) ECHAM5-wiso. The panels (e) to (g) present values between 40-70°N for (e) GNIP data, (f) iLOVECLIM, (g) LMDZ4 and (h) ECHAM5-wiso. The horizontal line in the box plots corresponds to the median value.*

- l 478: more complex: is it really more complex than in LMDZ? Any reference to justify this assertion? The main difference between LMDZ and ECHAM seems to be the horizontal resolution, not the complexity of its parameterization.

We agree that the sentence was not clear. We changed it to: "*Among all three models and surprisingly, ECHAM5-wiso which least reproduces this continental effect, despite having a better horizontal resolution*".

- l 515: I don't understand the logic: why the absence of sea ice ... would lead to fractionation during sea ice formation?

The sentence was ill-formulated. In this iLOVECLIM simulation there is no sea ice and we observe high isotopic values in the polar ocean. However, if sea ice was taken into account, a fractionation would happen during sea ice formation, leading to depletion of the liquid water isotopic composition as explained in Werner et al. (2016). We modified the sentence in the revised manuscript.

- l 518: more depleted d-excess -> lower d-excess

Done.

- l 521: as well highly depleted -> very low

Done.

- General rule: the vapor is depleted, but the δ18O, the d-excess or the 17O-excess are low.

We replaced the word depleted by low, and enriched by high in the revised manuscript.

- l 544: observation -> observations or dataset

Done.

- l 560: than d-excess -> as the d-excess maximum

Done.

- l 573: with the conservation of -> with a reasonable simulation of

Done.

- l 579: that has a too important -> with an excessive

Done.

- l 580: isotopes-enabled -> isotope-enabled

Done.

- l 581: a good accordance of -> with good agreement

Done.

**References**

Dansgaard, W.: Stable isotopes in precipitation. Tellus, 16(4) :436–468, https://doi.org/10.3402/tellusa.v16i4.8993, 1964.

Dee, S., Noone, D., Buenning, N., Emile-Geay, J., and Zhou, Y.: SPEEDY-IER: A fast atmospheric GCM with water isotope physics, J. Geophys. Res.-Atmos., 120, 73–91, doi:10.1002/2014JD022194, 2015.

Moore, M., Kuang, Z., and Blossey, P. N.: A moisture budget perspective of the amount effect, Geophys. Res. Lett., 41, 1329– 1335, https://doi.org/10.1002/2013GL058302, 2014.

Risi, C., Muller, C., and Blossey, P.: Rain evaporation, snow melt, and entrainment at the heart of water vapor isotopic variations in the tropical troposphere, according to large-eddy simulations and a two-column model, J. Adv. Model. Earth Sy., 13, 1–28, https://doi.org/10.1029/2020MS002381, 2021.

Roche, D., Paillard, D., Ganopolski, A., and Hoffmann, G.: Oceanic oxygen-18 at the present day and LGM: equilibrium simulations with a coupled climate model of intermediate complexity, Earth Planet. Sc. Lett., 218, 317–330, doi:10.1016/S0012- 821x(03)00700-3, 2004.

Roche, D. M.: $\delta^{18}$O water isotope in the iLOVECLIM model (version 1.0) – Part 1: Implementation and verification, Geosci. Model Dev., 6, 1481–1491, doi:10.5194/gmd-6-1481-2013, 2013.

Werner, M., Haese, B., Xu, X., Zhang, X., Butzin, M., and Lohmann, G.: Glacial–interglacial changes in $H_2^{18}O$, HDO and deuterium excess – results from the fully coupled ECHAM5/MPI-OM Earth system model, Geosci. Model Dev., 9, 647–670, https://doi.org/10.5194/gmd-9-647-2016, 2016.

**Response to Reviewer 2**

We thank Reviewer 2 for the comments and suggestions that helped to improve the new manuscript. We addressed all the comments of the referee (responses in blue), including the addition of the new $\delta^{17}$O and $^{17}$O-excess data for a better model-data comparison, and changed the manuscript accordingly.

- Lines 13-14: "… and calculate the associated secondary markers d-excess and 17O-excess. So far, the latter was modeled only by the atmospheric model LMDZ4. Results of…"

We changed the sentence accordingly.

- Lines 136-137: It is not that true anymore. It is rather easy to run a snapshot simulation with an isotope-enable coupled model. However, you can say that the true challenge for coupled GCMs is the computing cost of transient simulations like the last deglaciation or the Holocene. Please change the sentence accordingly.

You are correct, thank you for highlighting it. The text now reads: "*General Circulation Models (GCMs) have first been used to simulate separately water isotopes in the atmospheric and oceanic components but are now capable of running snapshot coupled simulations with the water isotopes-enabled. Running transient coupled simulations like the last deglaciation or the Holocene remains however still challenging due to high computing cost of these GCMs*".

- Section 2.4: A very recent paper presenting a dataset of d17O has been released one month ago by Terzer-Wassmuth et al. from IAEA. I think the authors should include this dataset in their analyses. I know this requires extra work in terms of figure plotting and statistical analysis, and I'm sorry about that. But I think it's worth including this data set (or worth trying at least).

Thank you for pointing us this new dataset. It is indeed very interesting to add them to the manuscript. We added these new data in the revised manuscript, respectively to the Figures 1 ($^{17}$O-excess spatial distribution), 3 (Taylor diagram for $^{17}$O-excess), 6 ($^{17}$O-excess model-data comparison), A1 and A2 ($\delta^{17}$O spatial distribution and model-data comparison).

We revised the text in Section 3.1.3 considering the new data and said, as already highlighted in the previous manuscript, that we have a too high dispersion of the $^{17}$O-excess data in iLOVECLIM. Indeed, higher values than observations are modelled from mid to low latitudes and lower values than observations at high latitudes of the northern hemisphere. The Figure 6 in the manuscript has been changed including an additional zonal comparison between iLOVECLIM, LMDZ4 and the observation (see Figure 1 below). Interestingly, the opposite pattern in the models compared to observations suggests that the physical processes at play are not fully understood and require further investigation.

We added the following sentence in the Appendix A for the spatial distribution of the $\delta^{17}$O: "*Similarly, the values over land are lower than over the ocean. In comparison to the available data (including new data from Terzer-Wassmuth et al. 2023), iLOVECLIM calculates higher values of several permil in central Europe and Canada, and lower values in Africa. Agreements are observed between the model and the data in East Asia, western Europe and North America*".

[Figure]

Figure 1: (a) Relationship between the iLOVECLIM modelled isotopic value and $^{17}$O-excess measurements, without values in Antarctica. LMDZ4 model results are also presented. The regression curves between model and data are presented in dark blue for iLOVECLIM and red for LMDZ4 with the confidence bands. The 1:1 line are shown with the black dashed lines. The errors bars associated with the data are shown at 1σ. (b) Zonal $^{17}$O-excess comparison. The model results (in color) are compared to observations (in grey). The different lines are polynomial regression curves for the model results that co-locate with the observations.

Finally, we modified the model-data $\delta^{17}O$ comparison text in the Appendix A to reflect the new data that have been added to the Appendix Figure.

- Line 324: "Despite these biases, iLOVECLIM reproduces the global trend of depleted…"

Done.

- Line 479-480: Replace the sentence by something simpler like "More generally, iLOVECLIM models too high d-excess values from mid- to low-latitudes (Figure 2b)"

Done.

- Lines 484-486: "In Figure 2b, we excluded these outlier values for a more suitable model intercomparison. Zonal mean d-excess values from mid- to high-latitudes modeled by LMDZ4, GISS, and CAM are too high compared to the observations, whereas values from ECHAM5-wiso are systematically too depleted."

We changed the sentence following your comment.

- Line 492: "(0.34 to 0.52), but with a higher SD compared to…"

Done.

- Line 529: Figure 4

Done.

- Line 552: Figure 5

Done.

- Line 650: Figure 6

Done.

- Line 658-659: data are now available from Terzer-Wassmuth et al. and in the WISER portal.

We added these new $^{17}O$-excess data to the figure for comparison with iLOVECLIM. We however only present data for Ankara and Reykjavik stations since the data are not available for Pretoria and Belem. Due to the limited number of measurements, the monthly mean $^{17}O$-excess is presented only for 2 calendar years within the period 2015-2018. Similarly to what has been done in the original manuscript, we normalized the values because the seasonal evolution of precipitation and isotopic ratio in the model is not expected to perfectly reflect the measurements. The model-data agreement is not perfect, especially for Ankara, but the model is able to reproduce the seasonal variations as observed in the data for Reykjavik. The seasonal evolution of $^{17}O$-excess at Ankara and Reykjavik (Figure 2) has been added to the figure in the revised manuscript in Section 3.1.4.

[Figure]

Figure 2: Monthly evolution of the $^{17}$O-excess at Ankara and Reykjavik. The red line is the GNIP data measured at the station and the blue line is the iLOVECLIM model result at the corresponding location. The data and model results have been normalized. The error bars for the data are also shown at 2σ.

- Lines 660-662: Why don't you take all the years of the simulation (100 years, as stated at lines 265-267) instead of of just 10 years?

We agree that starting from the equilibrium simulation, the additional 100 years for monthly outputs are as well at equilibrium. There is indeed not that much differences between monthly results for the whole simulation in comparison to the last 10 years as seen in Figure 3 for the precipitation and δ$^2$H$_{precipitation}$ at the same stations than presented in the manuscript.

[Figure]

Figure 3: Monthly evolution of precipitation (top) and δ$^2$H$_{precipitation}$ (bottom) at several stations (different columns for Pretoria, Belem, Ankara and Reykjavik). The two curves correspond to the mean over the last 10 years of the simulation (dark blue) and to the mean values over the entire simulation (light blue).

Based on that observation, we kept the last 10 years of the simulations for the monthly results.

- Line 694: Figure 7

Done.

- Line 728: Figure 8

Done.

- Line 732: "examining" instead of "looking at"?

The change has been done.

- Lines 736-737: Therefore, we selected for each GNIP station the pixel that was in better agreement with the precipitation and isotopic composition seasonal cycle data.

We corrected the sentence.

- Lines 741-742: This amount effect is -0.085‰/cm.y-1 in iLOVECLIM, weaker than the observed one in GNIP data (-0.139‰/cm.y-1).

We modified the sentence in the revised manuscript.

- Line 744: as already noted by Risi et al. (2010).

Done.

- Line 746: Figure 9

Done.

- Line 750: Temperature plays an important role on the hydrogen…

This has been changed in the text.

- Lines 759-760: Differences in modeled d2Hprecipitation between iLOVECLIM and LMDZ4 are enhanced for the lower values, and model-data agreement is deteriorated.

We changed the sentence accordingly.

- Line 760: What do you mean by "the difference in simulating the isotopic composition at low temperature"? Could you be more specific?

From the Figure 10 we can see that the modelled $\delta^2H_{precipitation}$ in iLOVECLIM presents higher values than LMDZ4 below a temperature of -20°C. But according to your last comment, we changed the sentence in the revised manuscript.

- Line 762: Figure 10

Done.

- Continental effect: Why do you focus only on tropics (0-20°N and 0-20°S)?

That is a good point. Please see the response of the same comment from Reviewer 1 above.

- Line 797: Figure 11

Done.

- Line 804: too enriched compared to observations at high latitudes.

Done.